# Interpretable Bike-Sharing Activity Prediction with a Temporal Fusion Transformer to Unveil Influential Factors: A Case Study in Hamburg, Germany

Sebastian Rühmann [1,*], Stephan Leible [2] and Tom Lewandowski [2]

1   Department of Computer Science, Human-Computer Interaction, University of Hamburg, 22527 Hamburg, Germany

2   Department of Computer Science, IT-Management and -Consulting, University of Hamburg, 22527 Hamburg, Germany; stephan.leible@uni-hamburg.de (S.L.); tom.lewandowski@uni-hamburg.de (T.L.)

*   Correspondence: sebastian.ruehmann@studium.uni-hamburg.de

**Abstract:** Bike-sharing systems (BSS) have emerged as an increasingly important form of transportation in smart cities, playing a pivotal role in the evolving landscape of urban mobility. As cities worldwide strive to promote sustainable and efficient transportation options, BSS offer a flexible, eco-friendly alternative that complements traditional public transport systems. These systems, however, are complex and influenced by a myriad of endogenous and exogenous factors. This complexity poses challenges in predicting BSS activity and optimizing its usage and effectiveness. This study delves into the dynamics of the BSS in Hamburg, Germany, focusing on system stability and activity prediction. We propose an interpretable attention-based Temporal Fusion Transformer (TFT) model and compare its performance with the state-of-the-art Long Short-Term Memory (LSTM) model. The proposed TFT model outperforms the LSTM model with a 36.8% improvement in RMSE and overcomes current black-box models via interpretability. Via detailed analysis, key factors influencing bike-sharing activity, especially in terms of temporal and spatial contexts, are identified, examined, and evaluated. Based on the results, we propose interventions and a deployed TFT model that can improve the effectiveness of BSS. This research contributes to the evolving field of sustainable urban mobility via data analysis for data-informed decision-making.

**Keywords:** bike-sharing system; bike-sharing activity; demand prediction; machine learning; Temporal Fusion Transformer; Long Short-Term Memory

## 1. Introduction

Since its inception in 1990, bike-sharing systems (BSS) have gained widespread popularity in urban settings, significantly contributing to the shift towards non-motorized urban mobility. These systems are integral to the concept of smart cities, fostering social inclusion [1], reducing carbon emissions [2], and enhancing urban mobility [3]. Modern BSS, functioning as networks of intelligent vehicles, provide valuable real-time data on mobility and transport in smart cities via the Internet of Things (IoT) and advanced sensor technologies. These innovations include the recent adoption of geofencing for virtual BSS stations, as highlighted by Caggiani and Camporeale [4]. It enhances the management of bike-sharing services by defining specific zones for bike pick-up and return.

The significant role of BSS in augmenting city efficiency and its synergistic interaction with public transportation is supported by various studies [5–8], which have demonstrated an expansion in mobility networks and improved connectivity between BSS and public transport. Empirical findings also suggest that BSS contribute to reducing traffic congestion [9,10]. For instance, Cheng et al. [11] discovered a notable increase in bike-sharing usage in Washington, D.C., during public transport disruptions, with a 10.1% rise in central areas and an 11.4% increase in peripheral regions. Furthermore, Adnan et al. [12] concluded

that BSS in smaller cities effectively bridge the first or last-mile gap in public transportation networks. This issue arises when the distance between an individual's home and the nearest public transport stop, or between the public transport stop and their workplace, exceeds one kilometer.

BSS promote community bonding via their collective usage [13] and also mitigate concerns about bike theft via the use of shared bikes [14]. However, improving bike availability remains a critical challenge, as user acceptance depends heavily on the system's reliability. Surveys have shown that ease of access (e.g., pick-up and drop-off) and system familiarity are key determinants of user satisfaction [15,16]. In contrast, insufficient availability can reduce user retention, whereas reliable BSS ensure better bike utilization and long-term value. For effective bike-sharing, balancing supply and demand is a crucial sustainability metric for BSS [17].

Some research suggests that balanced station utilization can be achieved via infrastructure modifications, user incentives, and strategic bike distribution in order to maximize station usage and minimize downtime [18,19]. Still, identifying suitable intervention strategies remains challenging, particularly when the cause of the imbalance is unclear. Understanding the moderating factors in bike-sharing activity is therefore essential for predicting BSS performance and enhancing system stability.

The objective of our study is to identify and analyze the key factors influencing station-based bike-sharing, assess their relevance, and contextualize them within the broader mobility and social framework. Therefore, we used collected data from the BSS in the city of Hamburg in Germany as a case study.

Acknowledging the significance of key factors, the subsequent objective of this study is to devise a method for predicting BSS activity. Predicting BSS activity poses challenges for multiple reasons. Firstly, the factors correlated with BSS activity demonstrate considerable variability, introducing fluctuations in bike ridership. Secondly, various factors are interdependent, leading to multicollinearity. Unlike simple linear relationships, BSS activity also engages in nonlinear interactions. Thirdly, certain correlating attributes remain unknown until the future (e.g., meteorological), compounding the complexity of accurate prediction. This confluence of factors culminates in modeling inaccuracies and uncertainties surrounding BSS activity predictions. Machine learning (ML), as an advanced statistical approach, has demonstrated its proficiency in accurately generalizing across various challenges and addressing these, including multicollinearity [20], non-linear learning [21], and learning of subsequential variance [22].

Building upon this foundation, we utilized ML to develop predictive models for BSS activity, aiming to assist operators, users, and policymakers. As a result, our study introduces an interpretable attention-based Temporal Fusion Transformer (TFT) model, a novel approach in the realm of bike-sharing research. We compare its performance with the Long Short-Term Memory (LSTM) model, which is considered the state-of-the-art technology for predicting BSS activity. Altogether, our study addresses the following research questions (RQ):

**RQ1.** *What are the key factors influencing bike-sharing system activity in Hamburg, and how do these align with recent findings and trends globally?*

**RQ2.** *How does the performance of an interpretable TFT model compare with the existing state-of-the-art LSTM model in predicting bike-sharing system activity?*

As there is a high number of abbreviations, we want to draw attention to Table A1 in Appendix A, which contains a list for better understanding and readability of our study. The structure of the study is outlined as follows: Section 2 presents an overview of related literature and research on the relevant core concepts and technologies. Section 3 details our approach, the environment, and the timeframe of the case study, outlines the methodologies employed and explains the execution of a comparative ML model experiment involving the proposed TFT model. Thereby, Sections 3.1–3.4 delve into the significance of various factors influencing the observed BSS activity. Sections 3.5–3.9 describe the ML model and the

experiment's methodology. The ensuing Section 4 evaluates the influence of various factors on BSS activity in Hamburg. Section 5 presents the ML model experiment, which consists of a comparative performance evaluation and an interpretability analysis. In Section 6, we contextualize and discuss the key factors of BSS activity and performance of the TFT model, enumerating interventions deduced from the experimental findings. Finally, Section 7 succinctly summarizes the key findings of this paper, and Section 8 outlines potential future directions for research in this field.

## 2. Related Research

### 2.1. Factors Mediating Bike-Sharing Activity

Recent studies indicate that a variety of factors influence bike ridership [23]. Table 1 presents an array of determinants, factors, and expected impacts on BSS activity identified by Eren and Uz [23]. Within this framework that we used as groundwork, endogenous determinants have a direct interaction with BSS, whereas exogenous determinants influence BSS activity from external sources. Often, exogenous factors are immutable, unpredictable, and fall outside the scope of direct control. Given the spatio-temporal nature of BSS, each factor influences bike-sharing activity both spatially and temporally. The magnitude and nature of these impacts can be conceptualized as an interplay between spatial accessibility and temporal availability.

**Table 1.** A framework of scope, determinants, and factors influencing BSS activity [23].

| Scope | Determinants | Factors | Expected Impacts |
|---|---|---|---|
| Endogenous | BSS | Age of BSS | NC—for non-member |
| | | Number of docks on station | PC—high trip volume |
| | | Number of available bikes | PC |
| | | Station buffer distance | Mixed |
| | Built environment | Bike pathways | PC |
| | | Population density | SPC |
| | | Recreation | PC |
| | | Residence | PC—high trip volume |
| | | Commercial | PC |
| | Public transportation network | Proximity of station | NC |
| | | Number of stations | PC |
| | | Travel distance | NC |
| | | Use of smart cards | SPC |
| Exogeneous | Meteorological | Season | Winter: SNC, else: PC |
| | | Precipitation | NC |
| | | Wind speed | NC |
| | | Humidity | SNC |
| | | Temperature | PC |
| | Socio-demographic | Younger age (>16) | PC |
| | | Higher income | PC |
| | | Holding driving license | NC |
| | | Male gender | PC |
| | | COVID-19 pandemic | PC |

Legend: strongly positive correlation (SPC); positive correlation (PC); strongly negative correlation (SNC); negative correlation (NC).

Concerning station-related factors, the number of docking stations and available bikes have been shown to positively correlate with bike rental activity [14,24]. Research by Buck and Buehler [25] shows that the placement of bike stations near public transport hubs significantly enhances user engagement. Additionally, the frequency of bike usage is associated with areas with a high concentration of workplaces, food establishments, and universities [14]. The spatial distribution of bikes is also positively related to the urban public transport system [26,27]. Proximity to bus and metro stations, particularly within 500 m and 1.5 km, respectively, has been identified as crucial for the distribution of shared bikes. Furthermore, neighborhood characteristics such as population density, retail employment density, and median income correlate with bike rental activity [24]. However, a statistical analysis conducted in Houston identified that at least 32% of the variance in pick-up and drop-off locations remained unexplained by linear models, which were inadequate in capturing these complexities [28]. This limitation has necessitated the adoption of non-linear modeling techniques, such as ML, to discern critical variables amidst the intricate interdependencies existing among them.

Examining individual motivations, research suggests a distinction in bike-sharing usage between leisure and commuting purposes. Leisure trips during weekends tend to increase demand in the afternoon [29], while commuting trips on weekdays are more prevalent during peak working hours, typically involving short journeys from residential to commercial areas [30]. Meteorological conditions also affect cycling activity. Studies consistently show a negative correlation between rainy days and bike ridership [31], with heavy rainfall [32] and lower temperatures reducing usage [33]. Seasonally, bike ridership declines in winter and peaks in summer and autumn [33].

In addition, the COVID-19 pandemic and related measures have impacted BSS activity [34]. Bergantino et al. [35] report a significant increase in cycling during and after COVID-19 restrictions, including a 67% surge in demand for New York's bike-share program. Schwedhelm et al. [36] show a doubling of ridership in Chicago's program compared to 2019 during that time. Xin et al. [37] outline changes in ridership flow and its spatio-temporal distribution patterns in New York during the pandemic, observing a substantial negative effect on BSS stability. In contrast, Jiao et al. [38] did not find any adverse impact on the stability of BSS in Seoul.

### 2.2. Evolution of Prediction Models of Bike-Sharing Activity

There is an increasing interest in developing mobility prediction models, as highlighted by recent research [39]. The prediction of spatio-temporal activity in this domain is complicated and involves the mentioned endogenous factors of station-related and individual motivation, but also a variety of exogenous factors like socio-demographical and meteorological ones. Conventional supervised learning algorithms, such as decision trees, often struggle with this complexity.

In response, the exploration of artificial neural networks (ANNs), known for their ability to decipher complex linear and non-linear relationships, is gaining momentum. ANNs are also more adept at managing noise compared to traditional regression analysis, a finding supported by several studies in the field of bike-sharing activity prediction [40–42]. In particular, deep neural networks (DNNs) have demonstrated proficiency in capturing intricate relationships within datasets [43]. Their effectiveness, coupled with adaptability to large-scale data, presents a significant advantage, especially when considering the integration of expansive open datasets. These datasets, encompassing traffic, weather, and event data from smart cities, contribute to the development of digital twins. Digital twins are real-time virtual replicas of physical entities, systems, or processes updated via sensor data [44].

While various forms of recurrent neural networks (RNNs), as subtypes of DNNs, have been extensively studied in bike-sharing, there is a noticeable gap in the exploration of attention-based DNNs. For instance, a study by Lee and Ku [45] on a dual attention-based RNN demonstrated superior short-term bike-sharing prediction performance over

other DNNs. Similarly, research by Wang et al. [46] showed enhanced performance via the integration of attention mechanisms and convolutional neural networks. In traffic flow analysis, attention-based DNNs like the TFT outperformed both LSTM and Gated Recurrent Unit (GRU) models [47]. In comprehensive comparisons with alternative models, as shown by Lim et al. [48], the TFT model outperforms others in the BSS superordinate field of traffic. Specifically, it surpasses DeepAR by 69%, ARIMA by 135%, and ETS by 148% in terms of performance.

The architecture of TFT is particularly conducive to time series prediction, offering three pivotal advantages for the mobility sector. First, it adeptly incorporates future-unknown covariates and heterogeneous time series features, allowing for a more comprehensive consideration of relevant factors, which can lead to improved prediction accuracy. Second, its multi-horizon quantile forecasting provides detailed forecasts with quantified confidence levels. This feature allows operators, users, and policymakers to establish an acceptable range of uncertainty for the practical application of the prediction model. Third, the architecture of TFT offers interpretability, addressing the limitations of 'black-box' models. This transparency allows ML developers to verify the model's reliability over time and highlights opportunities for refinement and enhancement to practitioners.

## 3. Method and Data

In our study, we examined the trips of the BSS in Hamburg, Germany, over two years from 2021 to 2022. Since 2022, Hamburg has been designated as a federal model region for urban mobility [49]. The national aim is to innovate and establish a new, digitalized, and interconnected form of urban mobility by 2023 [50]. This focus makes Hamburg an especially pertinent location for mobility research, offering unique insights into the field. As an overview, Figure 1 shows our study design in three thematic blocks and the associated activities sequentially.

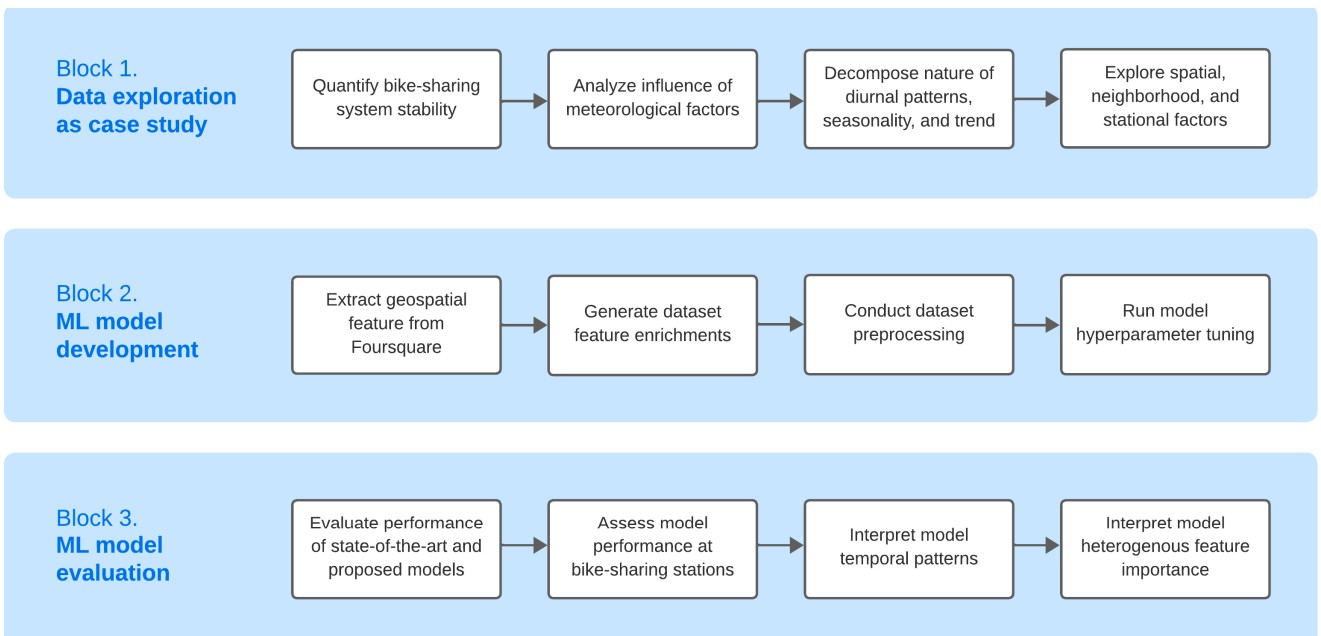

**Figure 1.** Study design and activities.

### 3.1. Study Area and Timeframe

The case study was conducted in the city of Hamburg, Germany, home to approximately 1.8 million residents. Known for its rich historical heritage and vibrant urban landscape, Hamburg is recognized as a significant port city and a pioneer in developing sustainable urban mobility solutions [51]. Among various transportation options, there is

a notable emphasis on promoting bike usage, aligned with the city's broader strategy to enhance the urban environment and foster sustainability [52].

In pursuit of advancing sustainable transportation, Hamburg has launched several initiatives. Our study builds on the Call a Bike program initiated by Deutsche Bahn [53] and operated in Hamburg since 2009. This BSS operates in conjunction with the public transport network, effectively integrating its infrastructure with bus, suburban train, and subway stations. This integration provides residents with an eco-friendly transportation alternative, enhancing connectivity and reducing reliance on motor vehicles.

The dataset for our study was sourced from the Deutsche Bahn, encompassing a critical period of two years: 2021 and 2022. This timeframe is particularly significant due to the global disruptions caused by the COVID-19 pandemic, which profoundly impacted human behaviors and mobility patterns. In Hamburg, various bike-sharing stations are strategically located at key public transportation hubs. Considering the city's extensive area and the numerous bike stations, our study categorizes specific zones of interest for a detailed analysis. Further details are described in the data exploration Section 4.2.

### 3.2. Datasets

The dataset for shared bikes comprises empirical booking data from all trips in 2021 and 2022. Each booking record includes a rental or pick-up timestamp, a start station ID, a return or drop-off timestamp, and an end station ID. Although the bikes are equipped with individual communication technologies, such as local radio, enabling individual tracking, the dataset does not contain bike IDs. The communication stack facilitates interaction with the base station for both pick-ups and drop-offs. Typically, shared bikes are secured with an external bolt at the docking station. When all docks are full, the system permits additional drop-offs in the vicinity of the base station. This infrastructure is maintained by Deutsche Bahn Connect (DB Connect) as part of the Call a Bike program [54]. The analyzed dataset includes a total of 3,639,055 records. A supplementary meteorological dataset encompasses various features sourced from OpenWeatherMap [55], including *temp*, *visibility*, *dew_point*, *feels_like*, *temp_min*, *temp_max*, *pressure*, *sea_level*, *grnd_level*, *humidity*, *wind_speed*, *wind_deg*, *wind_gust*, *rain_1h*, *rain_3h*, *snow_1h*, *snow_3h*, and *clouds_all*.

Furthermore, to encompass the social participation aspect of mobility, we extracted an additional feature. This involved gathering data on popular visitation hours to major attractions in Hamburg, identified as Points of Interest (PoIs). These PoIs include event venues, museums, and heavily frequented public transport stops, providing valuable insights into the patterns of social mobility within the city. In total, we compiled popular hours for 250 PoIs from the social network Foursquare [56]. Subsequently, we merged the PoIs based on their proximity to individual BSS stations. In cases where a PoI was close to multiple stations, it was associated with a few of them.

### 3.3. Bike-Sharing System Stability

The primary issue addressed in this research is the impediment to enhancing urban mobility and sustainability due to instability in BSS. To explore and quantify this aspect, Section 4.1 employs a tailored method that accounts for the unique activity patterns of each station. The data visualization is specifically designed to reflect these patterns over time. Initially, the dataset was organized by station, with bike-sharing pick-ups and drop-offs aggregated daily. Then, the average balance between pick-ups and drop-offs was calculated to quantify the BSS stability trend for each station as a metric value. Prior studies have underscored the significance of diurnal patterns in bike-sharing activities [57,58]. Consequently, our analysis also examines BSS stability throughout the day. For this purpose, pick-up and drop-off data were categorized by hour of the day and then averaged. The hourly balance was determined by subtracting the number of drop-offs from pick-ups per timestep, providing an accurate insight into the stability trend of each station.

### 3.4. Data Exploration and Factor Analysis

The volatility and complexity inherent in BSS data have been highlighted in similar studies focused on various cities, such as Melbourne and Brisbane in Australia [59], Fargo in the United States [60], and Marrakech in Morocco [61]. In our case study in Hamburg, we employ data visualization and statistical techniques to uncover characteristics, relationships, and initial patterns in the dataset. Additionally, we aim to contextualize the relationships within the ML model introduced in this study, tailored to Hamburg's unique environment, thereby facilitating reproducibility in future research.

Accordingly, in our analysis, we investigate the relationship of bike-sharing activity to meteorological, temporal, and spatial dimensions. As the sampling rate of the meteorological data was hourly, our resampled bike-sharing data matched up well with them. Initially, we explored the impact of meteorological changes on bike-sharing activity. Therefore, we employed pairwise Pearson correlation analysis. To ensure the reliability of the meteorological dataset, we replicated the experiment using precipitation data from the German Weather Service [62] of the observation station Hamburg-Fuhlsbüttel. Subsequently, we utilized a diagonal correlation heatmap matrix to display the pairwise correlations.

The temporal analysis investigates how bike-sharing activity concentrations change over time, focusing on the number of trips per weekday, hourly trends, seasonality, and long-term trends. Numerous studies have differentiated between the commuting motives of bike-sharing on weekdays and recreational use on weekends [23,63]. Hence, our data visualizations of bike-sharing activities are categorized accordingly. The analysis of seasonality and trends was conducted using Seasonal-Trend decomposition with Loess (STL), a method preferred over classical decomposition due to its enhanced accuracy, outlier reliability, and granularity. Unlike classical decomposition, STL employs locally fitted regression models to derive smooth estimates of the three components: seasonal, trend, and residual. We used a seasonal smoother of five and a trend smoother of 47 as parameters. The impact of COVID-19 measures on temporal patterns was integrated and evaluated using the ZPID Lockdown Measure Dataset for Germany [64].

In our study, we also conducted an examination of the spatial dimension of bike-sharing activity. By utilizing Python and packages such as Plotly and Mapbox, we developed a geographical map of Hamburg with BSS stations depicted as dot markers, positioned accurately by latitude and longitude. The level of BSS activity was computed based on the number of pick-ups and drop-offs at each station, with the balance calculated as the difference between these two metrics per hour. The code for our ML models and methods is provided as open source in the reference list [65].

### 3.5. Feature Selection

The judicious selection of features is crucial not only for the performance of the ML model but also for optimizing training time. The meta-study by Eren and Uz [23] highlighted the significance of meteorological factors in mediating BSS activity. Contrary to this, our factor analysis surprisingly did not identify a correlation between BSS activity and meteorological elements like rainfall, snowfall, and wind speed. To ensure that potentially beneficial non-linear relationships were not overlooked, we assessed the predictive value via the interpretability of TFT in various test models. The feature importance analysis revealed a minimal impact, leading us to exclude rainfall, snowfall, and wind speed from our experimental model. However, *temperature* and *humidity* exhibited a strong positive correlation and were consequently incorporated as features.

We also included known temporal factors to capture diurnal, weekly, and seasonal patterns. Specifically, the *is_weekday* feature aids in distinguishing between weekday commuting and weekend recreational activities. To facilitate the model's capacity for data extrapolation, a relative time index was introduced. Moreover, we applied data lagging to enhance the learning of seasonal characteristics. The optimal lagging values, determined in Section 4.2.2, are six (24 h), 42 (seven days), and 2190 (one year). Additionally, the dataset was enriched with logarithmic activity, average activity per station, activity scale,

and dynamic encoder length. Lastly, the PoI dataset was incorporated using a binary representation of *is_public_hour*, corresponding to the popular hours at these PoIs. An overview of all features of the ML models can be found in Table 2.

**Table 2.** All features of the ML models.

| Variable | Unit | Source | Type | Characteristic |
|---|---|---|---|---|
| *station* | Metric | DB Connect | Static, known | Categorical |
| *temperature* | °C | OpenWeatherMap | Time-varying, unknown | Continuous |
| *humidity* | % | OpenWeatherMap | Time-varying, unknown | Continuous |
| *is_public_hour* | Binary | Foursquare | Time-varying, known | Continuous |
| *activity* | Metric | DB Connect | Time-varying, unknown | Continuous |
| *activity_lagged_by_6* | Metric | DB Connect | Time-varying, unknown | Continuous |
| *activity_lagged_by_42* | Metric | DB Connect | Time-varying, unknown | Continuous |
| *activity_lagged_by_2190* | Metric | DB Connect | Time-varying, unknown | Continuous |
| *average_activity_by_station* | Metric | DB Connect | Time-varying, unknown | Continuous |
| *log_activity* | Metric | DB Connect | Time-varying, unknown | Continuous |
| *activity_scale* | Metric | DB Connect | Time-varying, unknown | Continuous |
| *weekday* | Metric | Computed value | Time-varying, unknown | Categorical |
| *is_weekend* | Binary | Computed value | Time-varying, known | Categorical |
| *time_of_day* | Metric | Computed value | Time-varying, known | Categorical |
| *month* | Metric | Computed value | Time-varying, known | Categorical |
| *time_idx* | Metric | Computed value | Time-varying, known | Continuous |
| *relative_time_idx* | Metric | Computed value | Time-varying, known | Continuous |
| *encoder_length* | Metric | Computed value | Time-varying, known | Continuous |

### 3.6. Indices of Performance

To accurately assess the predictive capabilities of the bike-sharing activity model, we considered various performance metrics. One key aspect was accounting for the frequent instances of zero values in bike-sharing activity, particularly during nighttime hours. Given that the Mean Absolute Percentage Error (MAPE) becomes infinite if actual values or predictions are zero, it was deemed unsuitable as a metric. Therefore, as indices of performance, we employed the Mean Absolute Error (*MAE*), Symmetric Mean Absolute Percentage Error (*sMAPE*), and Root-Mean-Square Error (*RMSE*). These metrics were chosen for their ability to meaningfully interpret the model's accuracy, even in the presence of zero values. The calculations for these metrics are detailed as follows:

$$MAE = \frac{1}{N}\sum_{i=1}^{N}|y_i - \hat{y}_i| \tag{1}$$

$$sMAPE = \frac{1}{N}\sum_{i=1}^{N}\frac{|y_i - \hat{y}_i|}{|y_i + y_i|/2} \tag{2}$$

$$RMSE = \sqrt{\sum_{i=1}^{N}\frac{(\hat{y}_i - y_i)^2}{N}} \tag{3}$$

Please note: Here, *N* represents the number of testing samples, *y* denotes the actual data, and *ŷ* refers to the corresponding prediction.

### 3.7. Individual ML Models

Bike-sharing activity prediction fundamentally relies on time series analysis. LSTM networks are widely utilized for predictions involving various types of sequential data, primarily due to their proficiency in capturing long-term dependencies. Thereby, in the realm of bike-sharing, several studies have demonstrated the superior performance of LSTM models in comparative experiments [42,66]. Recently, some novel studies have introduced other attention-based models [45,46]. However, LSTM continues to be the

prevalent state-of-the-art model for predicting bike-sharing activities due to its widespread acceptance and proven effectiveness. Therefore, in our study, we employed LSTM as a baseline for comparison against our proposed model, an interpretable attention-based TFT. This approach allowed us to evaluate the effectiveness of the TFT model in the context of established LSTM performance benchmarks.

### 3.7.1. Long Short-Term Memory (LSTM)

LSTM networks are an enhancement of RNNs specifically designed to capture long-term dependencies in sequential data. While standard RNNs suffer from two major issues, namely vanishing gradients and exploding gradients, which hinder their ability to process long sequences effectively, LSTMs are purposefully engineered to circumvent the long-term dependency problem [67].

The basic architecture of an LSTM includes a cell state and four interactive layers: an input gate, a forget gate, an output gate, and a cell update mechanism. These gates are instrumental in managing the information flow into and out of the cell state. This structure allows the network to maintain and utilize information over extended sequences, effectively addressing the long-term dependency issue. The cell state acts as the network's long-term memory, capturing dependencies across time intervals. Consequently, LSTMs have been extensively applied to time series prediction problems, demonstrating their usefulness in various application scenarios.

### 3.7.2. Temporal Fusion Transformer (TFT)

TFT is an interpretable DNN specifically designed for analyzing temporal dynamics and is adept at integrating heterogeneous features for multi-horizon forecasting [48]. The architecture of the TFT model comprises five main components: gating mechanisms, variable selection networks, static covariate encoders, temporal processing, and prediction intervals. The effectiveness of the model is fundamentally attributed to this intricate architectural design, which is illustrated in Figure 2.

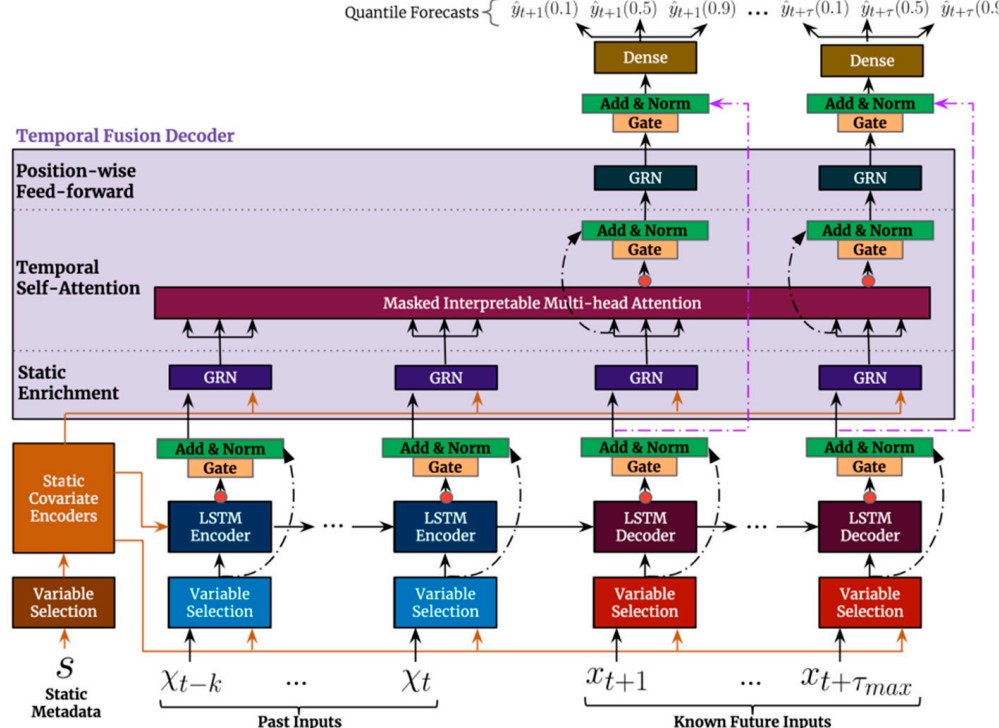

**Figure 2.** The TFT model architecture design [48].

A TFT implements time-series interpretability via multi-head attention mechanisms. These mechanisms pinpoint salient aspects of the input for each time step by defining the magnitude of attention weights. The orchestration of multiple attention heads facilitates the learning of diverse long-term temporal patterns. Additionally, a sequence-to-sequence layer is utilized to grasp the context of local patterns. As a result, the attention weights generated contribute to the model's comprehensive interpretability, enabling practitioners to detect anomalies and discern trends with the time series.

Furthermore, the TFT model incorporates type-exclusive variable selection networks. These networks select relevant input values for each time step while preserving the unique characteristics of the respective variable types. Each variable is assigned a selection weight, interpretable as an indicator of feature importance. TFT also effectively integrates static variables with temporal dynamics via context vector encoding. Continuous inputs are processed by an LSTM Encoder–Decoder mechanism, ensuring the integration of heterogeneous input types. Another distinctive feature of TFT is its ability to simultaneously predict various percentiles at each time step, enabling the generation of quantile forecasts and prediction intervals.

### 3.8. Data Preprocessing

To prepare for the ML models, the data underwent a preprocessing phase. The bike-sharing dataset was initially resampled to align with other datasets at an hourly frequency for data analysis. However, this dataset exhibited a low density for the nighttime hours due to reduced BSS activity. This sparsity could lead to an ML model bias skewed towards BSS idling values. To address this issue, the dataset was further resampled into four-hour intervals specifically for ML purposes, resulting in denser groupings and minimizing data distortion. Consequently, each day is represented by six intervals, with each interval encapsulating a four-hour interval.

In the process of aggregation, the datasets were adjusted to meet their respective empirical expectations. Meteorological features and PoI values were averaged, temporal features were computed based on their initial occurrence, and the BSS activity values were summed. Given the limitations of a singular LSTM model in handling future-unknown covariates, certain features such as *average_activity_by_station*, *log_activity*, *humidity*, and *temperature* had to be excluded.

### 3.9. ML Model Training Procedure

We partitioned the dataset into three subsets: training, test, and validation. During the training phase, each encoding sequence in the dataset was normalized using a scale individually fitted for it. This method of encoding normalization was chosen to prevent the look-ahead bias commonly induced by other normalization techniques. Before that, values were transformed using a softplus function to ensure non-negative inputs. A consistent seed was set and used to provide the highest level of reproducibility and comparability across all experimental trials.

Identifying the optimal hyperparameters was carried out using Optuna [68], which combined relative and independent sampling in an extensive study of 50 trials. To early stopping unproductive trials, a SuccessiveHalvingPruner was employed. Each trial was executed for a maximum of 50 epochs, with a limit of 30 training batches. Following this, the final model was trained on the entire span of training batches using the identified optimal hyperparameters. In the TFT model configuration, the learning rate was set to reduce by a factor of ten after a patience period of four epochs. The sampler in the proposed TFT model and compared LSTM model undertook a search across the ranges shown in Table 3.

The best configuration for the proposed TFT model and compared LSTM model was found with the hyperparameters displayed in Table 4.

**Table 3.** Search ranges of the sampler of the proposed TFT model and compared LSTM model.

| TFT Model | | LSTM Model | |
|---|---|---|---|
| **Parameter** | **Value Range** | **Parameter** | **Value Range** |
| Learning rate | 0.0001–0.1 | Learning rate | 0.0001–0.1 |
| Max. gradient norm | 0.01–1.0 | Max. gradient norm | 0.01–1.0 |
| Num. Heads | 1–4 | LSTM layers | 1–10 |
| Dropout rate | 0.1–0.5 | Dropout rate | 0.1–0.5 |
| State size | 8–320 | Hidden size | 8–128 |
| Hidden continuous size | 8–64 | | |

**Table 4.** Optimal configuration details for the proposed TFT model and compared LSTM model.

| TFT Model | | LSTM Model | |
|---|---|---|---|
| **Parameter** | **Value** | **Parameter** | **Value** |
| Minibatch size | 128 | Minibatch size | 128 |
| State size | 90 (15 days) | State size | 90 (15 days) |
| Tmax | 42 (7 days) | Tmax | 42 (7 days) |
| Learning rate | 0.001 | Learning rate | 0.06 |
| Max. gradient norm | 0.02 | Max. gradient norm | 0.02 |
| LSTM layers | 2 | LSTM layers | 1 |
| Dropout rate | 0.11493 | Dropout rate | 0.46659 |
| Hidden size | 51 | Hidden size | 79 |
| Hidden continuous size | 31 | | |
| Attention head size | 4 | | |

### 3.9.1. Loss Functions

The LSTM model underwent training using an MAE loss function (see Equation (1)), whereas the proposed TFT model was trained using a *QuantileLoss* function. This involved minimizing the cumulative loss across all quantile outputs.

$$QuantileLoss_\tau = \frac{1}{N} \sum_{i=1}^{N} \max(\tau(y_i - \hat{y}_i), (\tau - 1)(y_i - \hat{y}_i)) \tag{4}$$

When looking at the *QuantileLoss* formula, it is evident that the *MAE* loss is equivalent to the *QuantileLoss* when $\tau$ (tau) is set to 0.5. Therefore, in our study, the *QuantileLoss*, serving as the native loss function for the TFT model, remains comparable to the *MAE* loss function utilized in the LSTM model.

$$
\begin{aligned}
&QuantileLoss_\tau = \frac{1}{N} \sum_{i=1}^{N} \omega_\tau(y_i, \hat{y}_i) |y_i - \hat{y}_i| \\
&where: \\
&\omega_\tau(y_i, \hat{y}_i) \begin{cases} 1 - \tau & for\ y_i < \hat{y}_i \\ \tau & for\ y_i \geq \hat{y}_i \end{cases}
\end{aligned} \tag{5}
$$

### 3.9.2. Optimizer

Both models were trained using the Ranger Optimizer, as outlined by Wright and Demeure [69]. The Ranger Optimizer integrates the LookAhead and Rectified Adam (RAdam) techniques to enhance efficiency [70,71]. RAdam is known for its usefulness in stabilizing the initial phase of training, while LookAhead contributes to ongoing progress and supports convergence during training. Additionally, the Ranger Optimizer incorporates Gradient Centralization, a feature that can enhance the generalization performance of DNNs by regularizing weights and output features.

## 4. Results of the Case Study on the Bike-Sharing System in Hamburg

### 4.1. Stability Evaluation

The subsequent section delves into the key metric stability of BSS, which is pivotal in assessing the system's effectiveness. Effectiveness in this context is conceptualized as a zero-centered scale, where the optimal state is zero. A zero value indicates an ideal equilibrium between demand and supply at a station, whereas positive and negative values signify an overflow and underflow of shared bikes, respectively. Given that only bikes at well-positioned stations are likely to be rented, the stability of the BSS is inherently linked to its sustainability. Hence, enhancing stability is a direct route to improving the system's overall sustainability.

Figure 3 illustrates the stations in our case study that are underperforming in terms of BSS stability in 2021. The top and bottom five stations depicted in the figure exhibit a significant imbalance, with the former group showing a higher discrepancy in bike drop-offs (demand) and the latter group in bike pick-ups (supply). Notably, the *Hauptbahnhof / Heidi-Kabel-Platz* station (in the direct vicinity of the central train station) experiences the most substantial imbalance, with an average daily underflow of 11.27 bikes per day, indicating higher demand than supply. In contrast, the *S+U Landungsbrücken / Johannisbollwerk* station faces an overflow, with an excess supply over demand of 5.55 bikes per day. This imbalance is not merely transient but persists over a significant period, suggesting that its cause extends beyond exogenous factors like weather and socio-demographic motives. We infer that these imbalances are predominantly due to endogenous factors. A more detailed view of the issues caused by endogenous factors is presented from a spatial perspective in Section 4.2.3.

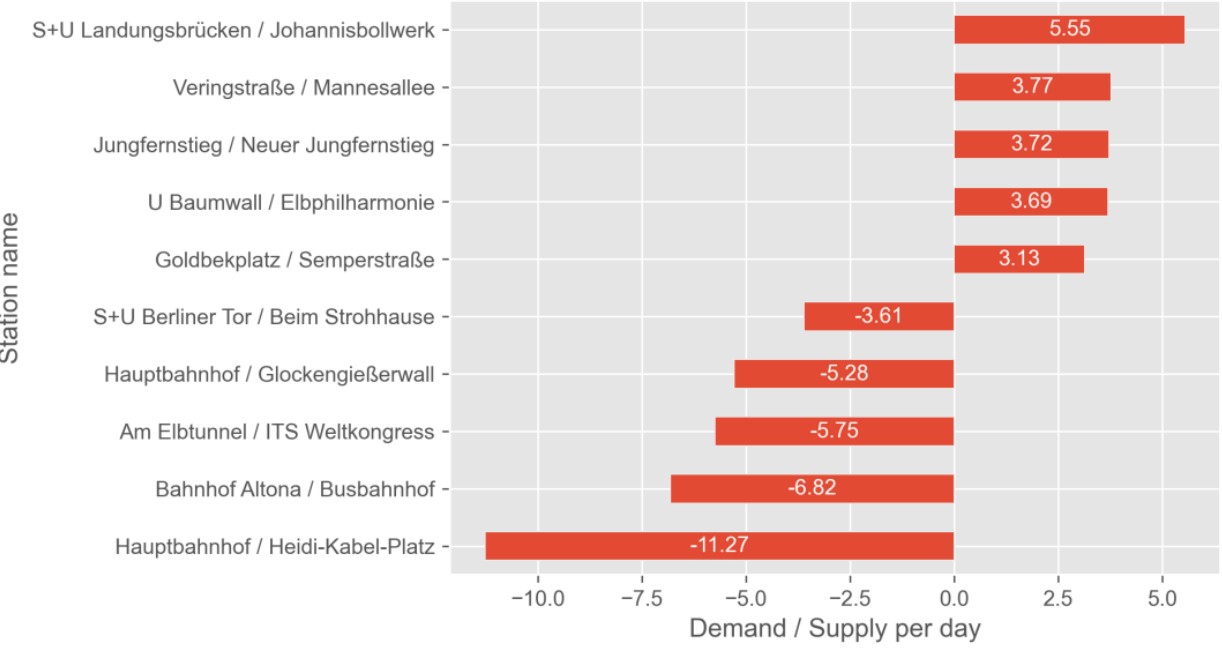

**Figure 3.** Average daily underflow/overflow of bikes at the top and bottom five stations in 2021.

Figure 4 depicts the imbalance between shared bike pick-ups and drop-offs at *Goldbekplatz / Semperstraße* throughout the day. This analysis, averaged hourly, reveals that until 14:00 (24 h clock), the demand for bike trips exceeds the supply. From 14:00 to 22:00, however, the supply surpasses the demand, as indicated by the blue line. Additionally, *Goldbekplatz / Semperstraße* ranks as the fifth-highest overflow station, as shown in Figure 3. Located in a residential area and equipped with 28 docks, it is considered a medium-sized

station. Typically, residential docking stations experience a morning peak in demand and an evening rise in supply, correlating with commuting patterns.

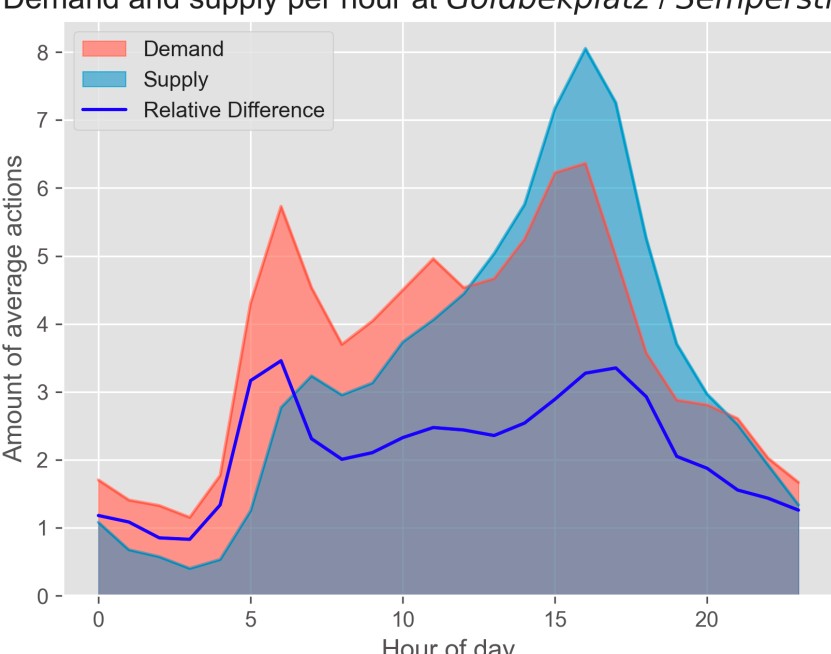

**Figure 4.** Average daily pick-ups and drop-offs per hour at *Goldbekplatz/Semperstraße* station.

The increased demand observed from 15:00 to 18:00, however, extends beyond typical commuting times. The station's vicinity offers convenient access to various amenities like pharmacies, cafes, and supermarkets frequented by both customers and commuting employees. Notably, amenities such as cafes and shops often remain open past regular working hours, with 15:00 to 18:00 marking a common period for the end of the workday. While this sustained high demand between 6:00 and 18:00 deviates from standard commuter traffic, it could be influenced by further endogenous factors like social interaction, as well as recreational and commercial interests.

Understanding all social influences on BSS activity is inherently complex. Nevertheless, as this observation aligns with existing research, it is advisable for operators, policymakers, and urban planners to thoroughly consider all related aspects affecting BSS stability within a temporal context [72,73].

### 4.2. Data Analysis

This subsection situates the proposed ML model within the specific socio-political and geographic context of the dataset used in this case study. Due to unexpected findings, the first subsection is specifically designated to meteorological factors. Further, we conducted a comprehensive examination of the dataset, focusing primarily on various factors from a temporal and spatial perspective. The analysis encompasses standard practices such as factor analysis, seasonality, and trend analysis, as well as exploratory analyses specific to the domain of bike-sharing. The findings from this section are particularly relevant and insightful for mobility researchers, policymakers, and operational stakeholders.

#### 4.2.1. Meteorological Factors

The relationship between meteorological factors and BSS activity from our case study is depicted in Figure 5. Among all the meteorological factors examined, only temperature and humidity showed a significant correlation with bike-sharing activity, with coefficients of 0.48 and −0.49, respectively. Contrary to findings in several existing studies that report a strong negative correlation between wind, rainfall, and snowfall with bike-sharing activity

(e.g., Refs. [72,74,75]), our case study did not find a correlation here. Other research suggests that BSS activity returns to normal after the experience of heavy precipitation after three hours (e.g., Ref. [32]).

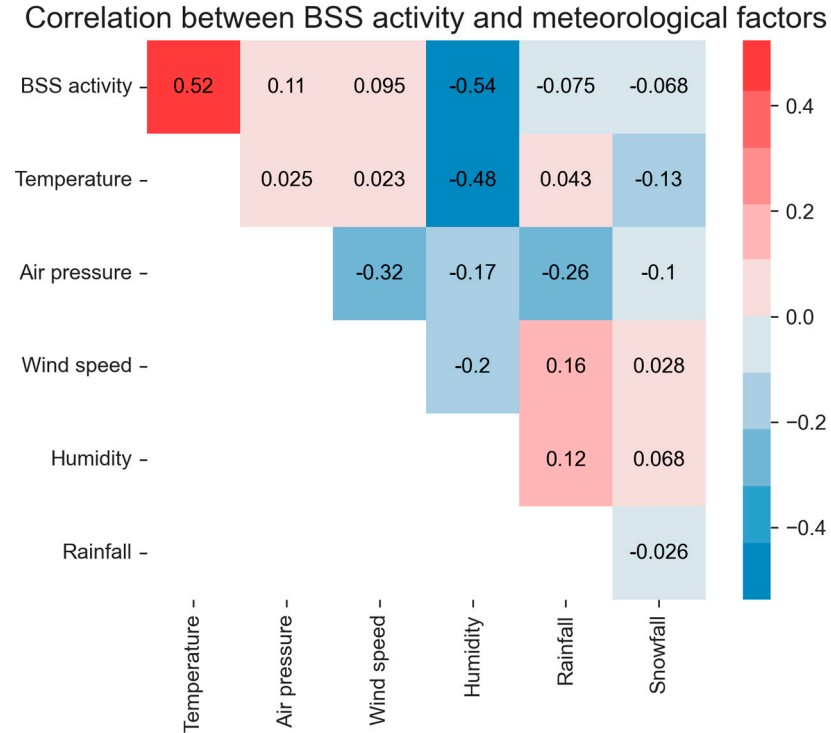

**Figure 5.** Correlation heatmap matrix of meteorological factors and bike-sharing activity.

Consequently, we undertook two approaches: (1) assessing the impact of anticipated rainfall by adjusting the rainfall feature's timing in our dataset and (2) repeating the experiment with an alternative precipitation data source. However, no correlation was observed, even when the rainfall feature was shifted by three hours earlier or later. The lack of correlation also persisted when correlating BSS activity with a different precipitation data source. Thus, contrary to our expectations, our results indicate no empirical evidence of a relationship between bike-sharing activity and wind speed, rainfall, or snowfall in Hamburg. One study states that cycling is less deterrent to precipitation in places accustomed to rain and snow [76], which could apply to Hamburg, as wind speed and rainfall frequently occur. Figure A1 in Appendix B includes boxplots that provide a detailed overview of the dataset used in this study.

### 4.2.2. Temporal Perspective

Figure 6 illustrates the seasonal variation in bike-sharing activity, with an uptrend during the warmer spring and summer months and a decline in autumn and winter. This pattern aligns with the observed correlation between temperatures and bike-sharing activity. Peak activity in bike-sharing during summer corresponds to the lowest idle times, whereas in autumn and winter, as the activity diminishes, idle times increase. Research indicates that bike-sharing activity is more sensitive to lower temperatures than higher ones (e.g., Ref. [33]).

In Hamburg, average temperatures in summer typically range from 20 to 23 °C, with occasional highs reaching up to 39 °C. Autumn sees a drop to an average of 12 °C, and winter daytime temperatures average around 5 °C, dipping to 0 °C at night. Despite Hamburg's relatively mild climate, with an annual average temperature of 13.6 °C, the dataset reveals a strong correlation between BSS activity, temperature, and seasonality.

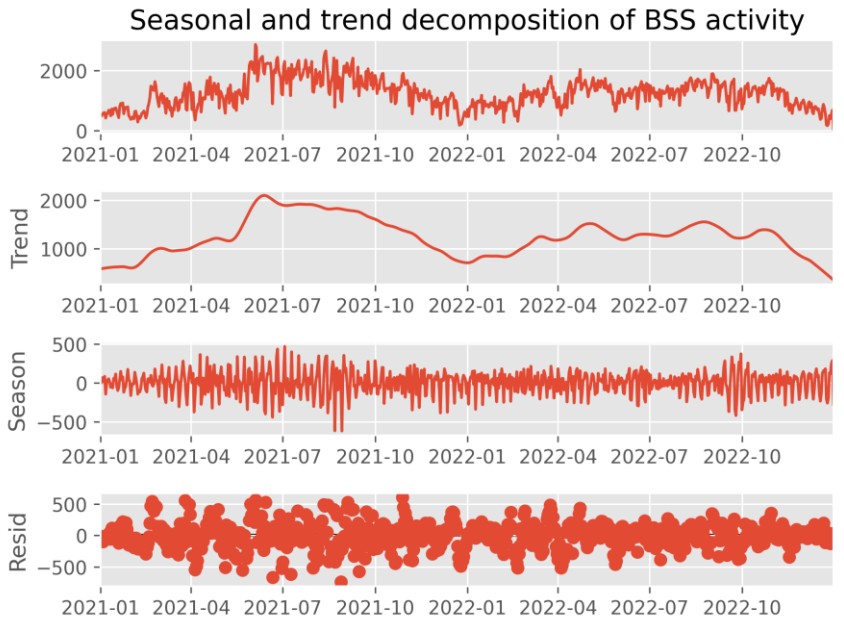

**Figure 6.** Temporal analysis histogram using STL.

The trend chart also sheds light on annual usage patterns. Notably, total BSS activity declined from 2021 to 2022. This decrease may be partly attributed to the social distancing measures implemented during the COVID-19 pandemic. After lifting these measures in October 2021, an average of 231 were booked daily. However, in the corresponding period of the following year (January 2022 to October 2022), the average booking rate dropped to 206 bikes per day. Contrary to Bergantino et al. [35], who reported a sustained increase in bike-sharing activity post-COVID-19 measures, Hamburg experienced a significant decrease. The seasonal component isolates seasonal fluctuations by excluding the trend, thereby normalizing the data to a baseline that omits long-term variations in BSS activity. Notably, the data reveal enhanced volatility in the seasonal aspect prior to September 2021, in contrast to the corresponding period in 2022. This variation may be ascribed to changes in COVID-19 measures.

Table 5 presents the total number of trips for each weekday, revealing that the activity from Tuesday to Saturday is relatively uniform, exhibiting a monotonous pattern. Notably, Monday shows a significant decrease in demand, with only 490,011 trips recorded. The data does not exhibit any distinct monthly trends. Demand reaches its lowest point on Sundays, which can be attributed to the nationwide closure of shops and supermarkets in Germany, resulting in reduced mobility needs. Our study reveals an unexplained reduction of 7.76% in trip count on Mondays compared to other weekdays, a notable discrepancy that remains unaccounted for within the scope of our research. To our knowledge, this anomaly has not been reported in other research.

**Table 5.** Accumulated BSS activity from our case study by weekday.

| Weekday | Number of Trips |
| --- | --- |
| Monday | 490,011 |
| Tuesday | 538,949 |
| Wednesday | 544,717 |
| Thursday | 527,477 |
| Friday | 555,043 |
| Saturday | 541,778 |
| Sunday | 441,079 |

The temporal distribution of trips throughout the day varies considerably between workdays and weekends. As previously mentioned, the number of trips on workdays and

Saturdays, representing a weekend day, is comparable. Figure 7 illustrates that the overall higher percentage of trips on workdays is due to there being five workdays compared to only two weekend days. During the weekend, activity begins to rise from 08:00, reaching a peak at 13:00, and then gradually declines until 22:00. Notably, the decrease in activity during the evening is more gradual, resulting in sustained bike-sharing activity levels.

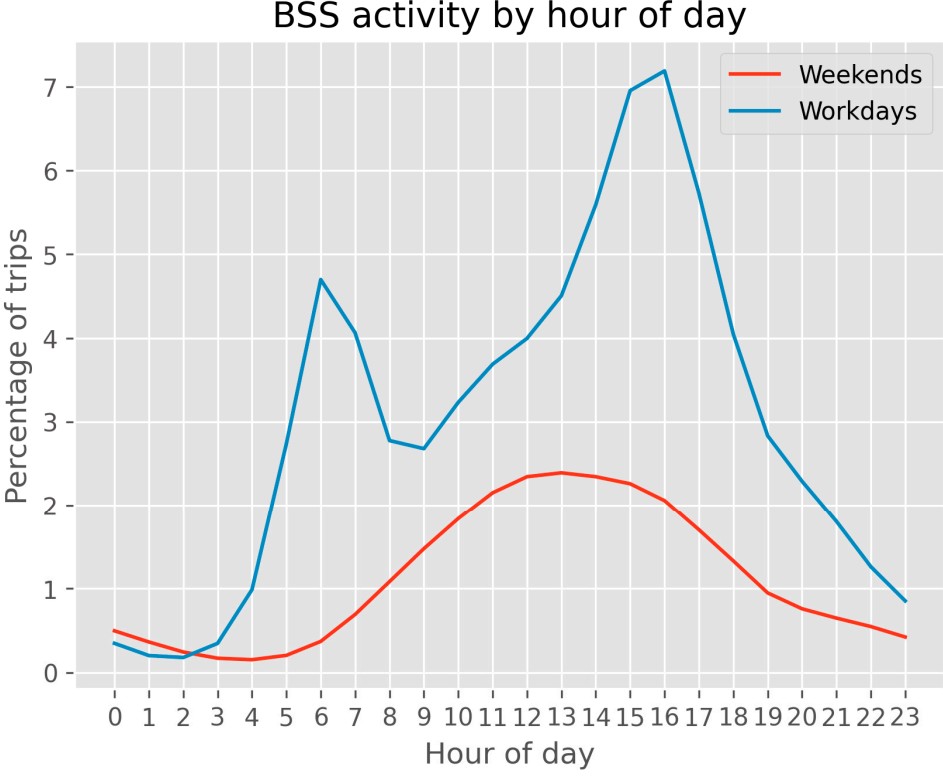

**Figure 7.** BSS activity by hour of day, categorized by workdays and weekends.

Conversely, on workdays, there is a noticeable increase in demand starting from 03:00, which then plateaus at around 06:00. This is followed by a decline over the next three hours until 09:00. From this point, activity rises almost linearly until reaching its daily peak at 17:00, after which it sharply decreases until 19:00 and continues to drop steadily until midnight. The hours from 00:00 to 03:00 are characterized by minimal bike-sharing activity, with less than 0.3% usage.

These observations align with studies that highlight distinct usage patterns between workdays and weekends (e.g., [77]). The data corroborate the notion that weekends excite more leisure trips, with peaks in activity during typical leisure hours. Additionally, the morning and evening peaks on workdays support the hypothesis that BSS are predominantly used for commuting during these times.

4.2.3. Spatial Perspective

Numerous studies have underscored the importance of proximity to surrounding stations in a BSS influencing bike-sharing activity. These studies cover a range of aspects, including the built environment [78,79], neighborhood [80,81], and the availability of public transportation [82,83]. In Figure 8, stations of our case study are depicted as circles, where the circle size reflects the level of bike-sharing activity; larger circles indicate higher activity, while smaller circles represent lower activity. The circle color denotes the balance degree, with white symbolizing perfect balance, blue suggesting a tendency towards overflow, and red indicating a propensity for underflow, as the legend on the right displays.

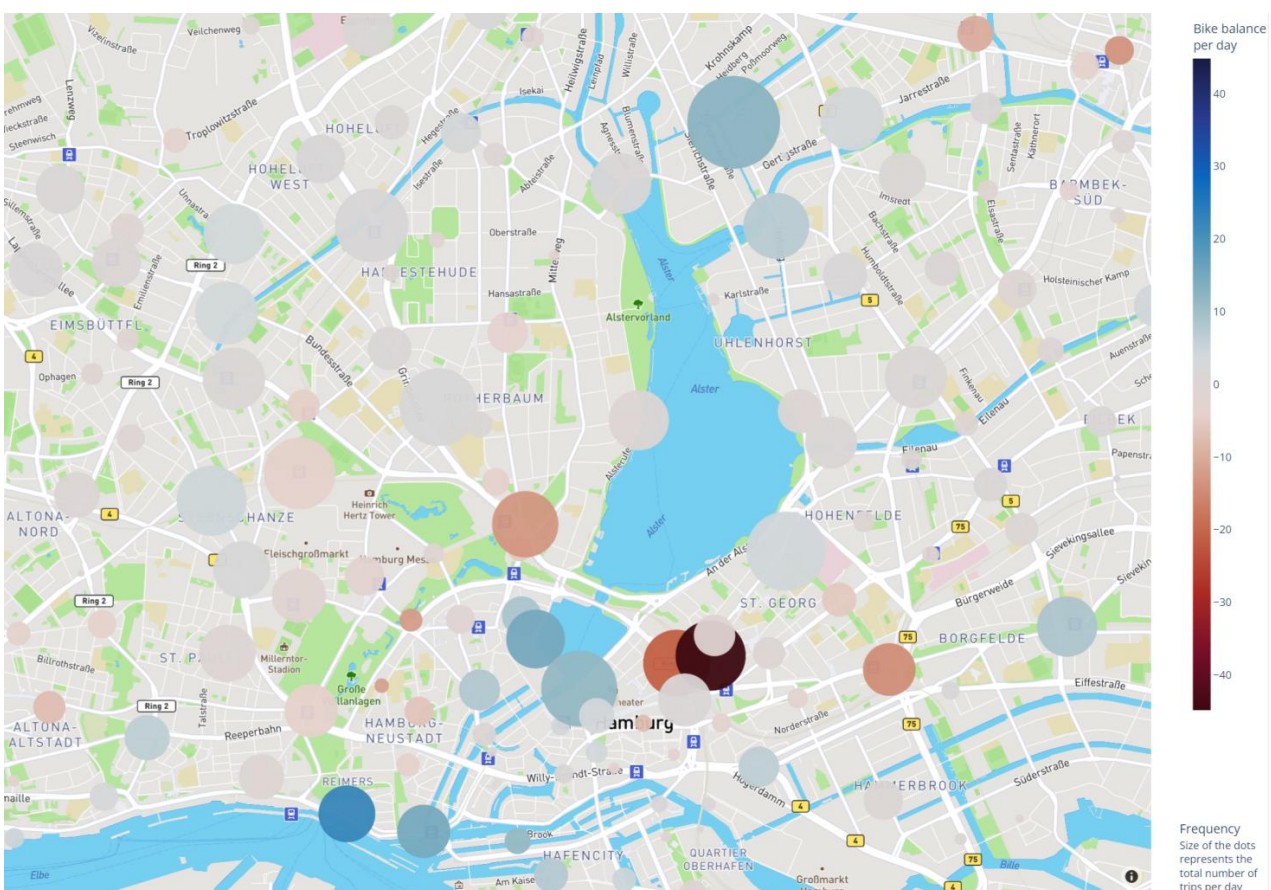

**Figure 8.** Combined spatial BSS activity and balance analysis. Description: the size of the dots visualizes the popularity of a station; the heatmap colors represent the BSS balance per day. An interactive version is available at https://seb.astian.eu/bss_activity_stability_map.html (accessed on 25 March 2024).

Spatially, the analysis reveals underflow at major Hamburg train stations like *Altona*, *Dammtor*, and *Hauptbahnhof*, all of which are significant transit hubs with regional and national connections. Similarly, underflow is observed at busy public transport stations with multiple lines, such as *U+S Barmbek* and *U+S Berliner Tor*. These patterns imply that shared bikes are more frequently used for departing from rather than arriving at these busy stations. Conversely, two clusters and some scattered stations exhibit high overflow: *Landungsbrücken*, *U Baumwall*, and, to a lesser degree, *Auf dem Sande* und *Am Kaiserkai* in the Hafencity district. The second cluster comprises *S+U Jungfernstieg*, *Neuer Jungfernstieg*, *Gustav-Mahler-Platz*, and *Axel-Springer-Platz*, which are predominantly characterized as recreational areas. The first cluster is located in the harbor area, popular among tourists and locals for its attractions, while the second is a touristic zone near the town hall, old town, and several upscale shops adjacent to the Alster (a tributary in Hamburg).

The *Goldbekplatz* station, near Stadtpark (a large park), a well-frequented green space with a planetarium, also shows high activity. This area, poorly serviced by public transport, experiences significant congestion during rush hours with limited car parking, resulting in high bike-sharing activity. Many bus routes are intended to meet the transportation needs of the area that is not served by rail. Referring to Table 1, the combination of a challenging public transportation network, a well-established built environment, and strategically located BSS stations can account for the heightened levels of bike-sharing activity observed. The proximity of *Goldbekplatz* and nearby amenities like supermarkets, shops, and cafes offer recreational opportunities, which emphasize the contribution of high overflow to the endogenous factor of recreation. As outlined in Section 4.2.2, focusing on bike-sharing

activity from a temporal perspective, this station consistently experiences a higher level of pick-ups and an increased number of drop-offs in the evening. The brief duration when high drop-offs lead to overall overflow suggests a scarcity of alternatives to road transport. In this context, BSS play a crucial role in alleviating congestion and reducing the need for parking.

*U Burgstraße* also exhibits a slight overflow, yet it is not situated in a recreational area, leaving the cause of this overflow unexplained from a spatial perspective. Meanwhile, a few other stations display high levels of balance, with most stations showing little or no imbalance. Therefore, it can be inferred that bike-sharing imbalance is not a widespread issue across the entire infrastructure. Rather, it is localized in specific areas, often following a repetitive temporal pattern. Comparatively, the issue of imbalance is more pronounced in terms of overflow quantity at stations in a BSS, whereas the degree of imbalance is more acute in stations experiencing underflow.

## 5. Machine Learning Model Experiment

This section details the evaluation of the ML model used in our study. We employed the TFT model in the context of bike-sharing and compared its performance against the state-of-the-art LSTM model. Both the TFT and LSTM models were developed using Python 3.8. The computational analysis was conducted on a system equipped with an Intel(R) Xeon(R) Silver 4110 CPU @ 2.10 GHz, 84 GB RAM, and four Nvidia GeForce GTX 1080 Ti 11 GB graphics cards, running on Ubuntu 20.04.6 LTS (Focal Fossa).

### 5.1. Performance Evaluation

Table 6 presents the performance outcomes for both the TFT and LSTM models. The TFT model demonstrated superior performance over the LSTM model across all metrics. Specifically, the TFT model achieved a remarkable 29.8% improvement in MAE, registering 0.98 compared to the LSTM's 1.40. Furthermore, the RMSE for the TFT model was 1.51, indicating a reliable approximation of the model's predictions. This RMSE value represents a 36.8% improvement over the LSTM model, showcasing the TFT model's enhanced capability in minimizing error variance. The sMAPE of the TFT model was 0.90, signifying a 17.5% improvement compared to the LSTM model. Overall, the TFT model not only reduced the total error but also decreased the variance of the error.

**Table 6.** Performance evaluation of the proposed TFT model and comparison of the LSTM model. Description: The best value and percentage indicating the improvement are shown in bold.

| Model | RMSE | MAE | sMAPE |
|---|---|---|---|
| TFT | **1.514 (36.8%)** | **0.9824 (29.8%)** | **0.9069 (17.5%)** |
| LSTM | 2.397 | 1.401 | 1.099 |

Our experiment design is comparable to that of Roussel et al. [84], who used the same dataset and similar preprocessing techniques. Their study, which employed a decision-tree model with PoIs as a predictive factor, achieved an RMSE of 3.2. As predictive models for BSS activity in Hamburg have not been extensively explored in other research, further comparative and independent analysis of our model's performance is limited. Subsequently, the predictions of the TFT model across different stations are underscored by common patterns identified throughout all stations. Each time index in the following figures corresponds to a four-hour interval.

Figure 9 shows the BSS activity prediction of our case study by the TFT model at a representative station selected for its typical past observations among all samples. However, the observed MAE loss of 0.55 at this station is lower than the overall average MAE of 0.9824 across all samples. The gray line in the figure indicates the model's attention weight for each timestep, following significant patterns from past timesteps. The ground truth values in the prediction timeframe show only minimal divergence from the learned patterns, leading to the model's above-average accuracy in this instance.

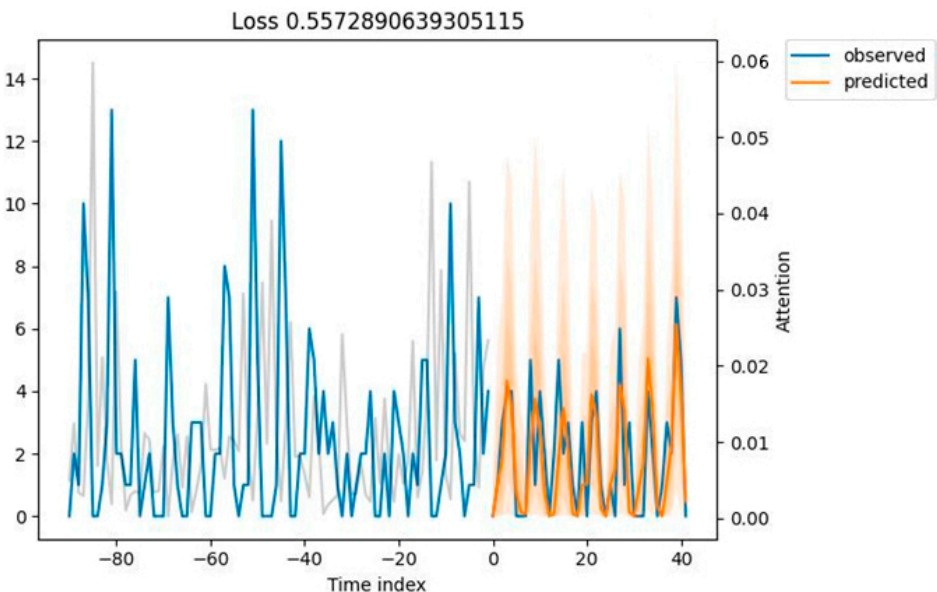

**Figure 9.** Visualized prediction on exemplary station #1 of the BSS. Description: the blue and orange lines represent BSS activity, plotted against the left axis; attention metrics are annotated in gray on the right axis. The array of quantiles is illustrated in varying shades of orange, while the MAE loss is prominently displayed in the chart's title.

Figure 10 presents a pattern in another station where past inputs closely mirror future predictions. Alongside the standard level of BSS activity observed in previous timesteps, there is a notable spike at the 30th timestep. Compared to the last sample, this instability, akin to autocorrelation, results in a less accurate approximation and a higher prediction error. Moreover, the model tends to overestimate BSS activity in both samples.

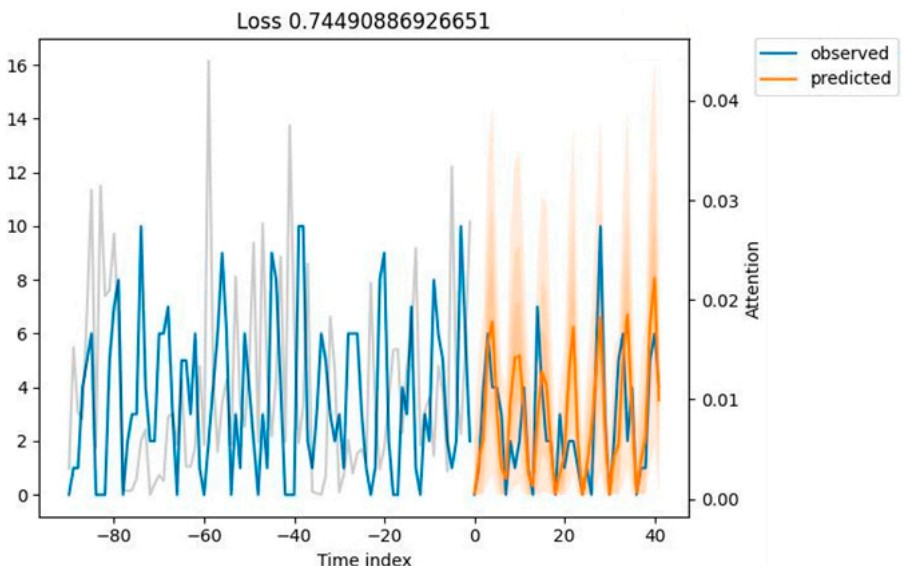

**Figure 10.** Visualized prediction on exemplary station #2 of the BSS. Description: the blue and orange lines represent BSS activity, plotted against the left axis; attention metrics are annotated in gray on the right axis. The array of quantiles is illustrated in varying shades of orange, while the MAE loss is prominently displayed in the chart's title.

The next two samples illustrate challenges faced by the proposed TFT model and offer insights for potential enhancements. Figure 11 shows a scenario with minimal overall BSS activity, where the model's predictions idle without convergence with the observations.

This indicates that low data expressiveness hinders accurate predictions, particularly at infrequently used stations. Despite this, the model maintains a low accuracy error of 0.159, which is negligible in terms of the model's general applicability.

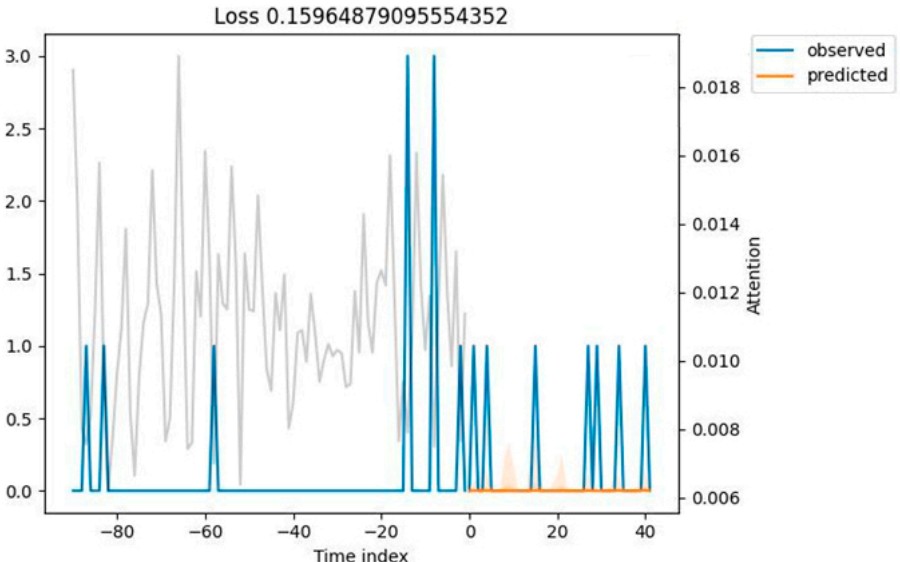

**Figure 11.** Visualized prediction on exemplary station #3 of the BSS. Description: the blue and orange lines represent BSS activity, plotted against the left axis; attention metrics are annotated in gray on the right axis. The array of quantiles is illustrated in varying shades of orange, while the MAE loss is prominently displayed in the chart's title.

Figure 12 exhibits the sample with the highest prediction error, at 4.729. In this instance, the model substantially underestimates the BSS activity. The actual activity in the prediction timeframe (>0th timestep) resembles the pattern observed before the −25th timestep. However, the model erroneously focuses on the downturn in the 25 subsequent timesteps. Although considering the most recent timesteps to be crucial for capturing progressive change may seem intuitive, in this particular sample, this approach leads to misguided predictions.

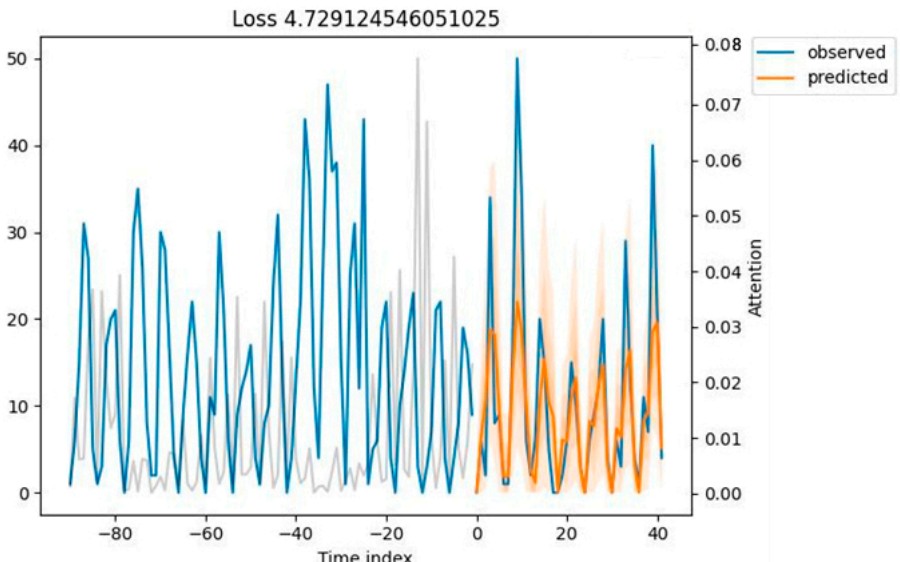

**Figure 12.** Visualized prediction on exemplary station #4 of the BSS. Description: the blue and orange lines represent BSS activity, plotted against the left axis; attention metrics are annotated in gray on the right axis. The array of quantiles is illustrated in varying shades of orange, while the MAE loss is prominently displayed in the chart's title.

Subsequently, Figure 13 enumerates the 25 most accurate and inaccurate predictions made by the model for each sample. It is observed that the model consistently overestimates BSS activity across all samples. Furthermore, inaccurately predicted samples exhibited lower normalized actual BSS activity compared to those predicted with high accuracy. The values in the normalized distribution indicate that the model's predictions are most inaccurate in scenarios where bike-sharing idling leads to data sparsity.

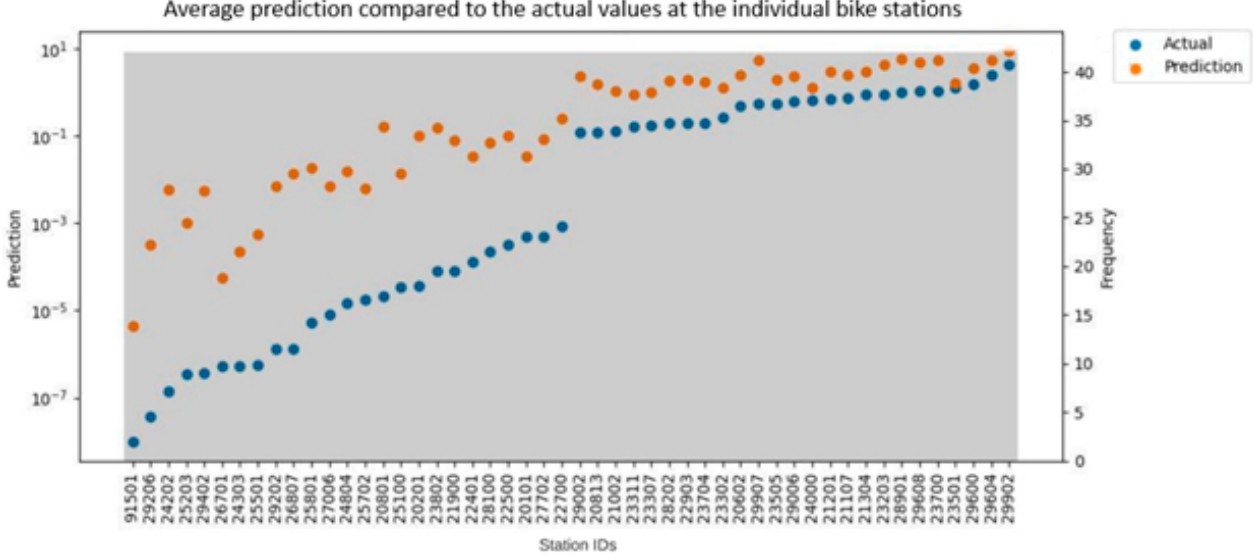

**Figure 13.** Normalized predictions versus actual values on 25 best and worst performances.

### 5.2. Model Interpretability

This subsection utilizes the distinctive interpretability afforded by the attention-based architecture of the TFT, which surpasses conventional DNNs. The analysis is structured into two subsections: (1) temporal patterns and (2) feature importance of static covariates, encoder variables, and decoder variables.

#### 5.2.1. Temporal Patterns

The self-attention mechanism in the TFT enables the model to focus simultaneously on information from different representational subspaces, thereby highlighting crucial features that are relevant at specific times. The TFT's multiple attention heads allow it to focus on multiple points in the time sequence, facilitating the learning of diverse long-term patterns. The implementation of TFT in this study involved integrating numerous heterogeneous inputs, leading to the creation of complex temporal and gated input context vectors. Based on an automated hyperparameter search, it was determined that the most effective TFT configuration for this complexity involves four attention heads, which is explained in further detail in Section 3.9.

Figure 14 shows the mean applied attention trend across all samples by timestep. The model intuitively prioritizes the most recent observations to inform the subsequent development towards the prediction timeframe. Notably, the TFT attention peaks at the $-15$th timestep, which is equivalent to 2.5 days before the prediction timeframe. Earlier, at 7 to 10 days before the prediction timeframe ($-42$nd to $-60$th timesteps), the attention maintains a sustained high level, emphasizing the significance of weekly patterns. Near the end of the encoding length, we observe several spikes of high importance before the 13-day timestep ($-78$th timestep). In summary, these various phases and the observed magnitude of attention in the context of recognized temporal patterns indicate the model's capability to learn multiple long-term patterns and local temporal contexts.

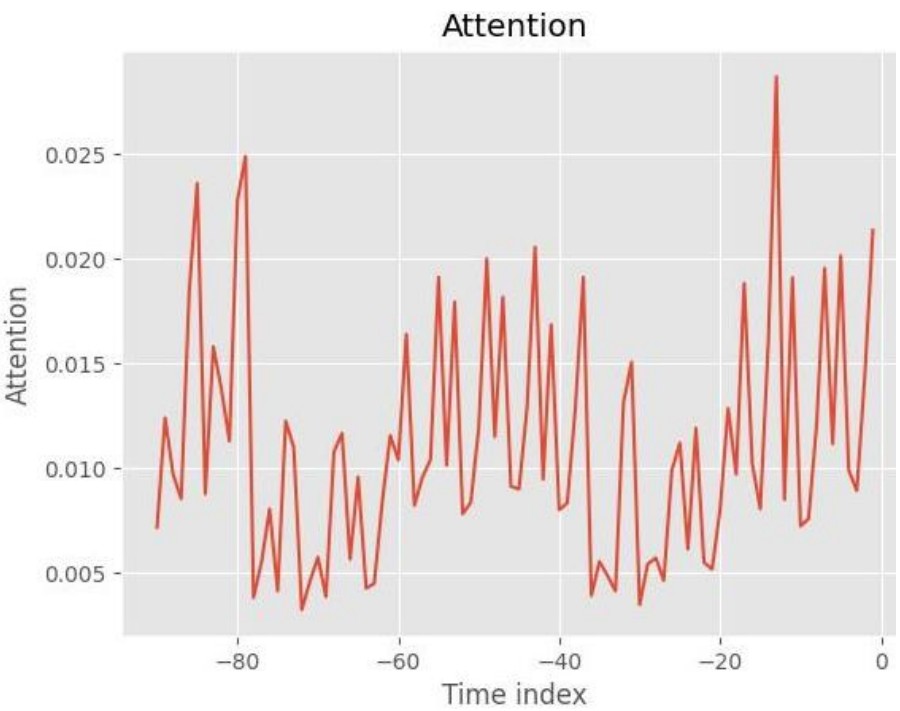

**Figure 14.** Average attention per timestep of the proposed TFT model.

## 5.2.2. Feature Importance

The feature importance within the TFT is determined by the selection weights computed by the Variable Selection Network (VSN) module. This module identifies the most salient features based on the input and modulates the information flow as a residual unit.

### Static Variables Importance

Static covariates, which are non-temporal external variables, can significantly enhance model accuracy when harmoniously integrated with the temporal dynamics of other inputs. In our experiment, the model incorporated four static covariates. Figure 15 underscores the notable importance of the unnormalized *activity_center* feature, a result of the dynamic normalization method employed. The EncoderNormalizer is tailored to fit each encoding sequence individually, thereby creating independently normalized encoding sets. The *activity_center* feature addresses the challenge of comparability between encoding sets by providing a consistent reference value across the normalized dataset.

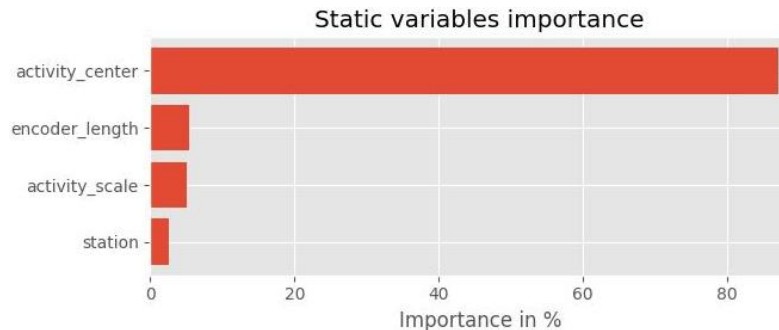

**Figure 15.** Average static variables importance of the proposed TFT model.

### Encoder Variables Importance

In the TFT, all non-static features function as encoder variables, which are crucial for generating a decoder hidden state representation. This hidden state is then used by the

decoder to extrapolate future unknown inputs, such as temperature (*temp*) and *humidity*. This underscores the vital role of pertinent encoder variables.

Figure 16 showcases the encoder variables and their respective importance as computed by the TFT model in this ML experiment. Notably, the model assigns the greatest importance to the *is_public_hours* feature, highlighting the link between popular hours at PoIs and BSS activity. The next three features—*activity_lagged_by_6*, *time_of_day*, and *is_weekend*—also demonstrate high importance, each accounting for approximately 11–13% of the model's attention.

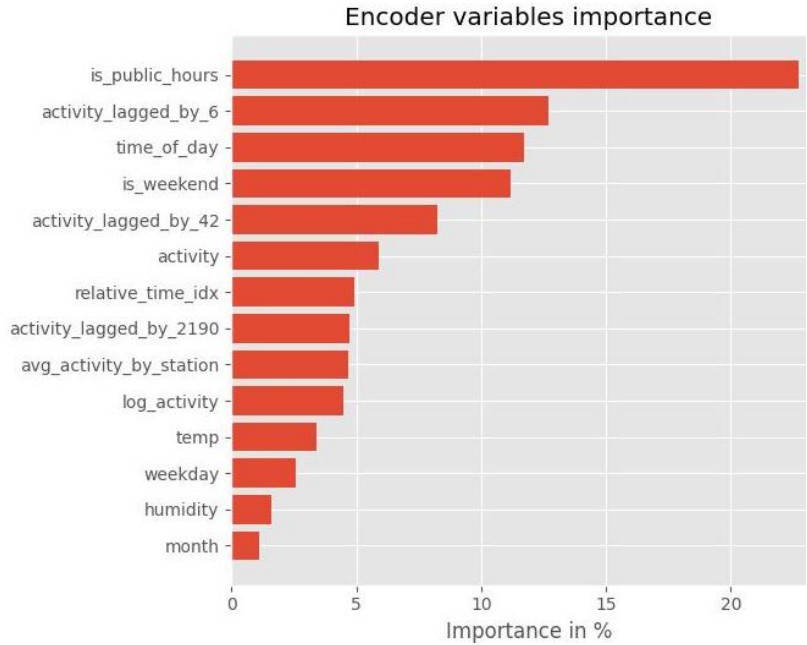

**Figure 16.** Average encoder variable importance of the proposed TFT model.

Among the three lagged BSS activity features, the diurnal (six timesteps) feature *activity_lagged_by_6* shows the most significant impact. The weekly lagged BSS activity also maintains moderate importance, surpassing regular BSS activity with a 7% relative importance. Subsequently, the features *relative_time_idx*, *log_activity*, and *avg_activity_by_station* each hold around 5% importance. Intriguingly, *relative_time_idx*—which represents the consecutive index within the encoding period—has, therefore, higher importance than meteorological features and the month. Given the typically ideal weather conditions during the summer prediction timeframe, we propose that the distribution of importance may vary for other episodes, influenced by the high variability of features about BSS activity.

Decoder Variables Importance

The decoder utilizes the hidden state representation to extrapolate encoder inputs and calculate future known decoder variables. These variables are enriched with static covariates and processed via temporal self-attention for making predictions. Essentially, decoder variables are those features for which future values are ascertainable at the time of prediction.

Figure 17 illustrates the relative importance of each decoder variable employed in this experiment. Notably, the *relative_time_idx* feature achieved the highest significance, accounting for 27% of the model's attention. This feature was also moderately important within the encoder variables, highlighting the significance of sequential order in encoding sequences. The *time_of_day* feature also received considerable emphasis, underscoring the importance of temporal dynamics. Remarkably, the *is_public_hours* feature, contributing 17% to the decoder variables and an additional 24% within the encoder variables, emerges as one of the most impactful features. This underscores the relevance of popular hours at

PoIs. Moreover, the combined importance of *time_of_day* and *is_public_hours* contributes a total of 38% to the model's effectiveness. This significant weight reflects the dynamic interplay between endogenous factors, represented by *is_public_hours*, within the temporal dynamics of *time_of_day*.

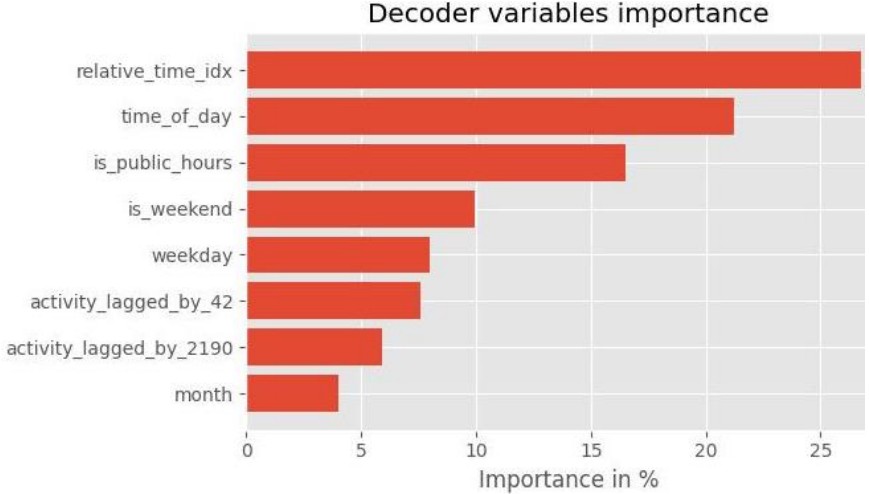

**Figure 17.** Average decoder variable importance of the proposed TFT model.

Furthermore, it is notable that *is_weekend* holds greater importance than the *weekday* feature, despite the latter's richer expression due to its higher granularity, representing all seven days a week. This could indicate multicollinearity, but it might also emphasize the efficacy of the binary *is_weekend* feature in distinguishing between BSS activity patterns during workdays and weekends.

## 6. Discussion

Surging BSS are an increasingly prominent form of sustainable transportation that reduces carbon emissions, expands the public transportation network, and promotes community bounding [4,5,11,13,82]. Thereby, the functioning of the BSS is impacted by various volatile heterogeneous endogenous and exogenous factors [8,23,29,38]. Data exploration and machine learning techniques with enforced interpretability can help to develop robust predictive algorithms (e.g., [39,43,66]), revealing the complex interplay of factors and ultimately creating inventions that support urban transportation problems, such as congestion, system failures, and instability.

This study contributes to this field via an in-depth analysis of the bike-sharing activity in the city of Hamburg, Germany, presenting a novel case study in terms of BSS factors and utilizing and explaining a predictive ML model. We focused on quantifying the stability of BSS, a critical aspect of the system's effectiveness and a key metric for its sustainability, as defined by Yahya [17]. Additionally, our research explored the underlying causes of BSS imbalances. In this context, we investigated the principal factors influencing bike-sharing activity from temporal and spatial perspectives. Furthermore, our study demonstrated that TFT, as a new proposed machine learning model in the realm of bike-sharing, significantly surpassed the performance of the existing state-of-the-art models while enabling full interpretability.

Theoretical and empirical research consistently illustrates that targeted interventions can enhance BSS stability [85–87]. However, a challenge is to select the right interventions and implement them appropriately. Inadequate interventions can inadvertently lead to further BSS instability, and neglecting to intervene may hinder progress in smart cities in general and mobility innovation in specific [88]. Our study suggests that special attention should be paid to specific contexts. In particular, we found a spatial correlation between underflow at BSS stations adjacent to busy train stations and overflow at BSS stations in recreational areas. These results should be considered when developing a city's BSS

and mobility network. Consequently, effective interventions should aim to increase bike drop-offs at busy train stations and bike pick-ups in recreational areas, especially during weekend peaks.

In the following, we discuss four types of interventions to improve BSS stability that serve as an inspiration for initial measures:

- Incentivization of shared-bike trips: Encouraging users to balance stations suffering from underflow or overflow can be an effective intervention strategy. While the current model of offering 30 min free of charge provides limited incentives, introduction rewards such as reservations for cargo bikes or pre-booking privileges could have a greater impact. These incentives could be integrated into existing apps to reward users for helping to meet BSS stability needs. The study of Fricker and Gast [89] highlights the exponential improvement potential of incentivization techniques.
- Manual rebalancing by operators: Efficient rebalancing, involving the timing, amount, and destination of bike redistribution, can maintain BSS balance. Studies on rebalancing algorithms provide valuable insights into this process [90]. Integrating predictions from the proposed TFT model can further refine the efficiency of these rebalancing interventions.
- Implementation of temporary pop-up stations in the BSS: Particularly during periods of increased activity, such as summer in recreational areas, temporary pop-up stations can help distribute BSS activity more evenly. Although the implementation of these stations can be challenging due to the need for IoT devices that are able to perform local radio communication and installed base stations, the advent of new-generation bikes equipped with GPS communication offers a promising solution. These advanced bikes can enable geofenced returns without the need for installed stations, simplifying the establishment of temporary pop-up stations. Thereby, pop-up stations can conceptually unite the unique advantages of free-floating BSS in addition to the present station-based BSS [91].
- Dynamic pricing models: A pricing model that varies according to temporal and spatial factors such as time of days, location, and demand is implemented. This includes offering discounts for picking up bikes from areas with an excess of bikes or for dropping off bikes in areas with an underflow [92]. In contrast, a slight premium is applied for renting bikes from areas of high demand or during peak hours. This strategy is designed to motivate users to naturally contribute to balancing the BSS.

By considering these interventions, operators and policymakers can enhance the functionality and sustainability of BSS, thereby contributing to the overall efficiency and evolution of urban mobility solutions. Furthermore, interventions can be specifically tailored to address particular issues:

- Offer discounted regional train tickets: Discounts on regional train tickets for those arriving at train stops via bike rides are provided. This approach is aimed at increasing the supply of bikes at busy train stations, thus addressing underflow issues at these locations.
- Relocate bike stations in a BSS closer to train boarding areas: The relocation of stations within a BSS closer to train boarding areas is undertaken to augment the unique advantage of the arrival with shared bikes. This move ensures improved connectivity over other transport modes, therefore offering a solution of bike-to-train connection for time-sensitive BSS users.

This study introduces a TFT model to predict bike-sharing activity on the basis of various heterogeneous factors. When compared to other ML models, TFT offers three key advantages for the application scenario of bike-sharing and BSS: (1) support for future-unknown covariates, (2) increased certainty, and (3) enhanced interpretability. However, it should be noted that the TFT model has higher (4) computational requirements.

1. Support for future-unknown covariates: The model can include inputs affecting bike-sharing that are typically uncertain in the future, such as meteorological and volatile traffic flow data, thereby improving model performance.
2. Certainty: By adding an array of quantiles to the model output, TFT enables reliable predictions for critical projects. This aspect is particularly valuable for long-term forecasting, essential in fields like urban planning, including BSS, as it provides a definitive probabilistic outcome.
3. Interpretability: The model sheds light on temporal patterns and feature importance, opening new avenues for feature engineering. It also allows BSS operators to identify significant changes in mobility activity, supporting the assessment of bike-sharing mobility adoption trends and the detection of anomalies, such as defects in the BSS infrastructure.
4. Computational requirements: The model features a more sophisticated architecture, encompassing numerous parameters, necessitating substantial hardware resources and computation time. These requirements are distributed across both model interfaces, affecting the interference and prediction times.

The selection of the most suitable model is greatly influenced by the specific application, its environment, and its prerequisites. In this context, Table 7 provides a compressed comparison, showcasing how these criteria are supported across various models.

**Table 7.** Compressed comparison of functional support by ML models. Description: The computational requirements range from the least demanding (denoted as "o") to the most demanding (indicated by "+++").

| Model | Future-Unknown Covariates | Certainty | Interpretability | Computational Requirements |
|---|---|---|---|---|
| TFT | X | X | X | +++ |
| LSTM | | | | + |
| DeepAR | | X | | ++ |
| ARIMA | | | | o |
| ETS | | X | | o |

The comparative analysis of the performance between the state-of-the-art LSTM model and the newly proposed TFT model demonstrated the superior prediction capabilities of the latter, achieving an accuracy of 1.51 RMSE compared to the LSTM model's 2.39. The accuracy of this approach exceeds the RMSE of 3.2 obtained using decision trees in a similar study [84]. The introduction of a multi-day forecasting window and the implementation of resampled four-hour intervals in the study's design enable this research to provide preliminary data for benchmarking purposes.

The level of accuracy renders the TFT model suitable for practical applications. Predictions of this model can support city planners and policymakers in enhancing bike infrastructure, strategically planning BSS stations, and facilitating the transition to sustainable transportation. Furthermore, these predictions can be integrated into navigation algorithms for BSS users, providing valuable estimates of bike availability at train stations. This feature is particularly relevant in the context of multimodal mobility, where adaptive navigation based on anticipated bike-sharing activity can ensure bike availability. Conversely, a lack of bike availability will likely result in user disengagement.

The deployment of accurate bike-sharing activity predictions supports data-informed decision-making for proactive BSS stability interventions. Consequently, these predictions can enhance BSS stability, leading to increased user retention and more efficient utilization of the distributed infrastructure. Collectively, these actions contribute significantly to the development of a resilient and sustainable mobility system.

This study also presents the interpretability of the TFT model, highlighting the relative importance of various factors affecting a BSS. These interpretability results offer insights for further enhancing model performance. The public hours of PoIs emerged as a significant feature in our model despite the simplicity of social participation modeling and feature

extraction used. In light of other research findings, refining this feature with more precise sampling and sophisticated feature extraction appears fruitful. A promising methodology for geospatial feature extraction tailored to bike-sharing activity in Hamburg is discussed in the study of Roussel et al. [84], offering a pathway for further development.

This research is not without limitations. One notable is its specific focus on the city of Hamburg, which may limit the generalizability of the findings as the key factors for BSS stability to other urban environments with different socio-economic dynamics, cultural factors, and infrastructure layouts. Therefore, the unique characteristics of Hamburg may not be representative of other cities. Consequently, the applicability of the derived interventions may vary when applied to different urban contexts with distinct bike-sharing patterns and user behaviors. Also, the configuration of the proposed TFT model specific to the dataset found in Hamburg should be carefully finetuned in other environments, which may end in diverging performance results.

Another constraint is the reliance on historical data patterns, which inherently assume that future behavior will mirror past trends. This assumption may not hold in scenarios where sudden changes occur, such as new urban development projects, changes in public transportation schedules and infrastructure, or significant shifts in user behavior due to external factors like pandemics or policy changes. However, this study shows that the novel interpretability stack can be utilized for reasoned model alignment.

## 7. Conclusions

In this study, we developed an interpretable TFT model to enhance the understanding of DNN performance in predicting bike-sharing activity. Additionally, the study conducted a thorough analysis, comparison, and contextualization of the key factors influencing bike-sharing activity, using Hamburg, Germany, as a novel case study.

The advancement of the TFT model compared to the LSTM model is attributed to the TFT model's built-in interpretability, which effectively addresses the limitations of traditional black-box models. In terms of local bike station activity prediction, the TFT model achieved an RMSE of 1.514, marking a 36.8% improvement over the LSTM model. The study's use of a naive geospatial public hours feature garnered high importance in the model, underscoring the critical role of PoIs in BSS activity modeling.

Regarding key factors, our findings indicate that humidity and temperature are significant predictors of BSS activity. However, in contrast to existing research that commonly reports a strong negative correlation, our study found no correlation between BSS activity and factors like snowfall, wind speed, and precipitation. This divergence from prevalent research findings suggests that the response to weather conditions in bike-sharing usage may vary across different geographical locations. Additionally, contrary to trends observed in other studies, bike-sharing activity in Hamburg did not sustain its post-COVID-19 levels but instead experienced a decline. This could potentially be influenced by initiatives like Germany's 9 Euro ticket. This highly affordable public transportation offer, introduced in 2022, provided unlimited travel on regional and local public transport across Germany. Such an attractive and cost-effective alternative to bike-sharing could have led many potential bike-sharing users to opt for public transportation, thus contributing to the observed decrease in bike-sharing activity in Hamburg during this period. This outcome highlights the potential impact of local factors and circumstances on bike-sharing usage patterns. Additionally, our analysis revealed a surprising 7.76% reduction in trip counts on Mondays compared to other weekdays.

In addressing BSS stability, we observed underflow at stations near busy train stations and overflow in recreational areas. Generally, the issue of imbalance is more pronounced with a higher occurrence of overflow than underflow at BSS stations, although the degree of imbalance is more acute in stations experiencing underflow.

## 8. Outlook

However, the findings of this study invite further exploration. Unlike the TFT model, LSTM lacks the capability to incorporate future inputs as unknown covariates, leading to the exclusion of some features. Future research could benchmark the TFT model with other (interpretable) attention-based models. Furthermore, exploring the application of the TFT model for real-time monitoring and dispatching in productive BSS operations could provide new valuable insights as well as enhance operational efficiency and user satisfaction. Expanding the analysis to include more datasets in comparative environments could improve results, particularly in scenarios with sparse BSS activity observations. Additionally, examining key factors by clustering stations could yield more precise station profiles. Future investigations might also clarify the unexpectedly absent correlation between bike-sharing activity and meteorological factors like rainfall, snowfall, and wind speed. Moreover, a comprehensive analysis of BSS activity beyond October 2021, once a substantial dataset covering an extended period has been compiled, may uncover novel usage patterns. This is particularly pertinent when examining the transition from the COVID-19 era to the post-COVID-19 period. Further studies should also delve into the cost–benefit analysis of different intervention types within BSS to ascertain their efficacy in promoting system stability. Understanding the economic and operational implications of these interventions will be critical in optimizing the BSS management strategies in smart cities and ensuring their sustainability in the urban mobility landscape.

Historically, the accessibility of bike-sharing data has spurred developments in mobility software. The novel case study and the TFT model presented here contribute to the broader goal of enhancing general mobility prediction, taking into account the diverse facets of sustainable urban mobility.

**Author Contributions:** Conceptualization, S.R. and S.L.; methodology, S.R. and S.L.; software, S.R.; validation, S.R.; formal analysis, S.R.; investigation, S.R.; resources, S.R. and S.L.; data curation, S.R.; writing—original draft preparation, S.R. and S.L.; writing—review and editing, S.L. and T.L.; visualization, S.R., S.L. and T.L.; supervision, S.L.; project administration, S.R. and S.L.; funding acquisition, S.R. All authors have read and agreed to the published version of the manuscript.

**Funding:** This research originates from the project *Ava-Citybike*, which received funding from the project *Digital and Data Literacy in Teaching Lab* and the *Stiftung Innovation in der Hochschullehre*.

**Institutional Review Board Statement:** Not applicable.

**Informed Consent Statement:** Not applicable.

**Data Availability Statement:** All data generated, extracted, and analyzed during this study that are not protected by copyright are included in this article.

**Acknowledgments:** The authors appreciate the effort and expertise of the Computer Center of the University Hamburg for providing computing resources for the machine learning experiment.

**Conflicts of Interest:** The authors declare no conflicts of interest.

## Appendix A. List of Abbreviations

**Table A1.** List of abbreviations of our study.

| Abbreviation | Definition |
| --- | --- |
| Adam | Adaptive Moment Estimation |
| ANN | Artificial Neural Network |
| ARIMA | Auto-Regressive Integrated Moving Average |
| BSS | Bike-Sharing System |
| COVID-19 | Coronavirus SARS-CoV-2 |
| DB | Deutsche Bahn |
| DNN | Deep Neural Network |

**Table A1.** *Cont.*

| Abbreviation | Definition |
| --- | --- |
| ETS | Error Trend and Seasonality, or exponential smoothing |
| GRU | Gated Recurrent Unit |
| LOESS | Locally estimated scatterplot smoothing |
| LSTM | Long Short-Term Memory |
| MAE | Mean Absolute Error |
| ML | Machine Learning |
| NC | Negative correlation |
| PC | Positive correlation |
| Radam | Rectified Adam |
| RMSE | Root Mean Square Error |
| RNN | Recurrent Neural Network |
| RQ | Research Question |
| sMAPE | Symmetrical Mean Absolute Percentage Error |
| SNC | Strong negative correlation |
| SPC | Strong positive correlation |
| STL | Seasonal-Trend decomposition using Loess |
| TFT | Temporal Fusion Transformer |
| VSN | Variable Selection Network |
| ZPID | Leibniz-Institut für Psychologie |

## Appendix B. Boxplots of Meteorological Data Distribution

Figure A1 depicts the distribution of four meteorological variables that unexpectedly exhibited no correlation with BSS activity.

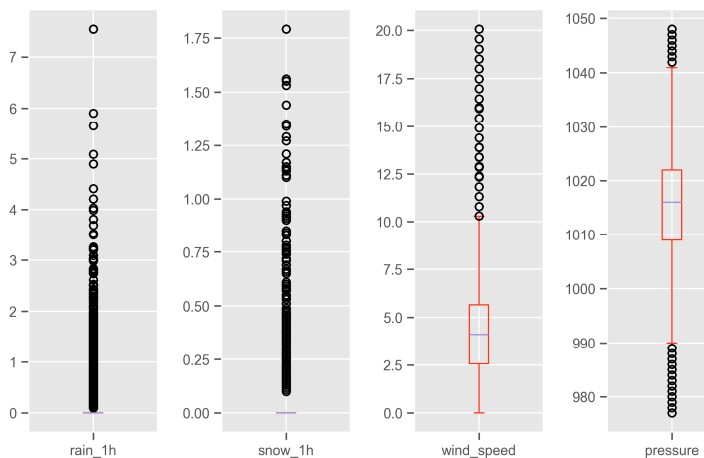

**Figure A1.** Boxplots of meteorological data distribution in our case study.

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
