# Peer review of "Interpretable Bike-Sharing Activity Prediction with a Temporal Fusion Transformer to Unveil Influential Factors: A Case Study in Hamburg, Germany"

_sustainability, doi:10.3390/su16083230_

Round 1

Reviewer 1 Report

Comments and Suggestions for Authors

Dear Authors,

There are aspects that are worthy of publication, however, this paper requires a major attention and careful considerations. Adequate revisions to the following points should be undertaken in order to justify recommendation for publication.

1. The advantages of the proposed method of this paper should be more highlighted.

2. Kindly enrich the literature review of this paper by citing additional references related to the topic addressed, particularly on influence of disturbances, modeling errors, various uncertainties in the described phenomenon/process. More generally, these reviews showcase the vibrancy of this field of research. In this sense, it could be the object of a brief consideration focused on the advances on the topic and make relation with these papers, which have to be discussed in Introduction section in the context of a more comprehensive literature review. I believe this would further strengthen the introduction and lend support to the methodology applied in general.

3. Comparison with different studies (models) should be provided widely in the Discussion section.

4. All references are provided applicable, however, I suggest extracting these key references (with other authors' approaches) from the text and putting them into a table focusing on weaknesses. Otherwise, it is very difficult to follow the Literature Review section.

5. In addition to that, the paper does not have the main points, which are necessary to make the manuscript more readable e.g. key notations or a list of abbreviations (in the table). It is very difficult to follow the figures, diagrams and text without the separate list of used nomenclature.

6. Please consider rebuilding Figure 5 to make numerical results more readable (maybe change font color).

7. Regarding the Figure 8. I see that you describe the size of dots, but the legend will be very useful. Please see Figure 9 in paper: Betkier, I., Macioszek E., 2022. Characteristics of Parking Lots Located along Main Roads in Terms of Cargo Truck Requirements: A Case Study of Poland. Sustainability 14(23), 15720. https://doi.org/10.3390/su142315720 and consider a similar approach.

To sum up, please analyze the above concerns. I recommend rebuilding the manuscript and taking into consideration my comments.

Regards,

Author Response

We want to thank you for the valuable feedback and the time dedicated to reviewing our submitted paper. The insightful comments and suggestions have significantly contributed to enhancing the quality and clarity of our research. We deeply appreciate your thorough analysis and constructive critiques, which have been instrumental in guiding our revisions and improving the overall rigor of our work.

Reviewer 1
ID Comment from reviewer Reply from authors
1 The advantages of the proposed method of this paper should be more highlighted. We have highlighted the advantages of the proposed and evaluated TFT model in several places in the article. Further, we have also added text parts on the motivation of our project and paper in Section 1 (Introduction) and additional information on the conceptual advantages in Section 2 (Related Work) as well as the advantages of the TFT model based on our test results in Section 6 (Discussion) and Section 7 (Conclusion). In a compressed form, we have included a comparison with other models in the new Table 7. We believe that in the main body of the article we consistently point out when the TFT model produces better or inferior results compared to the LSTM model.
2 Kindly enrich the literature review of this paper by citing additional references related to the topic addressed, particularly on influence of disturbances, modeling errors, various uncertainties in the described phenomenon/process. More generally, these reviews showcase the vibrancy of this field of research. In this sense, it could be the object of a brief consideration focused on the advances on the topic and make relation with these papers, which have to be discussed in Introduction section in the context of a more comprehensive literature review. I believe this would further strengthen the introduction and lend support to the methodology applied in general. In the Introduction, we have added more information on challenges and problems (multicollinearity, non-linear learning, and learning of subsequential variance) in the prediction of bike sharing activity and expanded it with new referenced literature [20-22]. This literature includes solution approaches to these challenges by machine learning. So our article provides an improved common thread for the motivation of our work and methodological approach with machine learning and the proposed TFT model.
3 Comparison with different studies (models) should be provided widely in the Discussion section. We have expanded the comparison alongside the new Table 7 in the Discussion. It compares in a compressed form the functionality support of popular temporal models. We have additionally incorporated a performance comparison of the Temporal Fusion Transformer (TFT) model within Section 2 (Related Work), utilizing a common benchmark dataset from the overarching field of traffic. This comparison underscores the performance advantages of the TFT model, as detailed in the cited paper (DOI). We have also added a recent and appropriate source from the Sustainability Journal (https://doi.org/10.3390/su16062569), which limits its finding on spatial factors with an unexplained variance. It prompts for neural network approaches on unveiling non-linear interpendencies in influential factors, underlining our methodology. 
4 All references are provided applicable, however, I suggest extracting these key references (with other authors' approaches) from the text and putting them into a table focusing on weaknesses. Otherwise, it is very difficult to follow the Literature Review section. The literature review section on mediating factors contains our key references, which are summarized in the scope, determinants and factors framework (see Table 1). At the beginning of Subsection 2.1, we present the relationship between the table and references. The subsequent Subsection 2.2 had previously tended to focus on ML and lost the BSS focus. Therefore, this section has been further specified to explain the specific benefits of the proposed TFT model for operators, users and policy makers in the BSS context. We address the weaknesses of other models indirectly by highlighting the functional advantages with the new Table 7 in the Discussion.
5 In addition to that, the paper does not have the main points, which are necessary to make the manuscript more readable e.g. key notations or a list of abbreviations (in the table). It is very difficult to follow the figures, diagrams and text without the separate list of used nomenclature. Done, we added a list of abbreviations as Appendix A. We also included a sentence in the introduction to draw attention to this newly added list for a better understanding and readability of the study.
6 Please consider rebuilding Figure 5 to make numerical results more readable (maybe change font color). Yes, improved readability with black font color
7 Regarding the Figure 8. I see that you describe the size of dots, but the legend will be very useful. Please see Figure 9 in paper: Betkier, I., Macioszek E., 2022. Characteristics of Parking Lots Located along Main Roads in Terms of Cargo Truck Requirements: A Case Study of Poland. Sustainability 14(23), 15720. https://doi.org/10.3390/su142315720 and consider a similar approach. We added a legend in Figure 8, which was a great hint for the readability of the figure. The attached Figure 9 from the Betkier & Macioszek (2022) article is in our opinion a very good approach, but unfortunately not transferable by us in the same way. We would have to generalize our data (metric BSS activity) to a categorical feature. Although this leads to a visually better illustration, it also loses its informative value, so when considering a limit of a few incrementally sized markers or fluidly sized markers, we prefer the current visualization.

Reviewer 2 Report

Comments and Suggestions for Authors

I loved reading the introduction, data description explanation of challenges additional to tabled and graphical displays for prediction.  For your future work, you may consider before and after COVID use of bike.  COVID changed subject's normal operation of daily activities.  Your model has explained the data but efficient and cost saving model building will be useful for prediction in the future

Author Response

We want to thank you for the feedback and the time dedicated to reviewing our submitted paper. The insightful suggestions have pointed out promising future research avenues.  

Reviewer 2
ID Comment from reviewer Reply from authors
8 For your future work, you may consider before and after COVID use of bike. COVID changed subject's normal operation of daily activities.   Yes, we added a sentence in the new Section 8 (Outlook) in which we discuss possible future work to examine a Pre- or Post-COVID-19 time to compare the results with ours during the COVID-19 time. It can be a very promising next study including behavior changes of people in cities during pandemics.
9 Your model has explained the data but efficient and cost saving model building will be useful for prediction in the future We have included the mention of a cost-benefit analysis in the new Section 8 (Outlook) of our study. We also believe that this is a very promising area for new research, but it is so complex that we can no longer include it in our study. However, this was not the aim of the work and requires new and more in-depth data in direct collaboration with local transport operators.

Reviewer 3 Report

Comments and Suggestions for Authors

Bike-sharing business research has placed technology and people's use(s) at the fore, where few consider its efficiency and sustainability. This article provides a wealth of practical, logical coherence and research argumentations regarding the ML (TFT and LSTM) models data and methodologies, BSS performance and enhancing system stability, and the frameworks for implementing customer-focused BSS activities and meteorological factors performance. Such interpretability of mobility prediction processes and conclusions identifies areas needing improvement and implementation steps at the value chain to transform urban mobility and bike-sharing ventures. 

A systematic and well-discussed essay.

Comments on the Quality of English Language

Some sentence structuring are needed to improve the texts, and misspelling to be corrected.  Apart from it, all is good to go.

Author Response

We want to thank you for the feedback that helped us to improve the English editing of our paper.
Reviewer 3
ID Comment from reviewer Reply from authors
10 Some sentence structuring are needed to improve the texts, and misspelling to be corrected. Apart from it, all is good to go. We have made some improvements by re-checking the article, the sentences and their structures as well as by implementing the comments of the other reviewers. We hope that this has improved the article's quality along with its readibility.

Reviewer 4 Report

Comments and Suggestions for Authors

Here are the key points, contributions, and future work of the paper listed:

 1. Explores the importance and challenges of bike-sharing systems (BSS) in smart cities. Focuses on studying the system stability and activity prediction of the BSS in Hamburg, Germany. Proposes an interpretable attention-based Temporal Fusion Transformer (TFT) model. Compares the performance of the TFT model with the state-of-the-art Long Short-Term Memory (LSTM) model. Analyzes the key factors influencing bike-sharing activity, especially temporal and spatial contexts.

2. The TFT model provides better interpretability, helping to understand the main factors affecting bike-sharing usage. Based on the analysis results, interventions, and an improved TFT model are proposed to enhance the effectiveness of the bike-sharing system. The research contributes to the development of sustainable urban transportation through data analysis and data-informed decision support and provides a data-driven approach for optimizing and decision-making in bike-sharing systems.

3. The study focused only on the bike-sharing system in Hamburg. Future work could validate the model's generalization capability using data from other cities. Future work could incorporate more potential factors affecting bike-sharing usage, such as weather, events, etc.

4. The study mainly focused on prediction and analysis. Future work could explore how to apply the model for real-time monitoring and dispatching in the actual operations of bike-sharing systems.

5. The paper should have discussed the specific costs and benefits of implementing the proposed interventions and deploying the model. Future work could conduct an economic cost-benefit analysis.

Author Response

We want to thank you for the feedback and the time dedicated to reviewing our submitted paper. The insightful suggestions have pointed out promising future research avenues.  

Reviewer 4
ID Comment from reviewer Reply from authors
11 The study mainly focused on prediction and analysis. Future work could explore how to apply the model for real-time monitoring and dispatching in the actual operations of bike-sharing systems. Done, this is an interesting idea and can result in valuable insights into the application of the model within a productive BSS system. We added this research opportunity in our future works paragraph.
12 The paper should have discussed the specific costs and benefits of implementing the proposed interventions and deploying the model. Future work could conduct an economic cost-benefit analysis. We have emphasized a little more that the intervention types are possible approaches that could be used are on the basis of other studies and the state of scientific knowledge. To get an impression here, we have added two additional references so that literature is available for each type of intervention. We also see the topic of costs and benefits as a logical next step, although its complexity and depth justifies a new paper.

With regard to our model, we have specified that the use of the model and the associated costs are significantly influenced by numerous factors, such as data, operating environment and available hardware.

As also noted by Reviewer 2, we have integrated the great potential of a cost-benefit analysis into the new Section 8 (Outlook) by adding a sentence to this topic motivating future work in this perspective.

Reviewer 5 Report

Comments and Suggestions for Authors

The article address an interesting topic, proposing a machine learning model for improving a bike sharing system.  The authors made an extensive study in the field, proven also by the large list of references. The study compares the proposed model, which is an attention-based TFT (temporal fusion transformers) model, with LSTM model. 

The paper is well written, but some aspects may be improved. Thus, the explanation of Figure 6 (lines 480-500) is not complete. It is obvious from the figure that the STL acts as a filter on the data shown in the first graph in the figure, but it would be better to have a brief explanation of how the STL works.

In figures 9-12, the left vertical axis represents quantiles? The value of a time step is 4 hours? I didn't find the time step value in the paper (maybe I missed it).

Author Response

We want to thank you for the valuable feedback and the time dedicated to reviewing our submitted paper. The suggestions have significantly contributed to enhancing the clarity of our research.

Reviewer 5
ID Comment from reviewer Reply from authors
13 The paper is well written, but some aspects may be improved. Thus, the explanation of Figure 6 (lines 480-500) is not complete. It is obvious from the figure that the STL acts as a filter on the data shown in the first graph in the figure, but it would be better to have a brief explanation of how the STL works. Done, an explanation of how the STL works is added before Figure 6.
14 In figures 9-12, the left vertical axis represents quantiles? The value of a time step is 4 hours? I didn't find the time step value in the paper (maybe I missed it). Done, we have added a basic introduction of the figures, which defines the time index with 4 hours. We also added a description to figures 9 to 12, which describe the diagrams.

Reviewer 6 Report

Comments and Suggestions for Authors

A list of comments from my end:

-why does TFT model better than LSTM model?

-many undefined parameters in each equation.

-provide a table for the notations.

-Most of the equations have been expressed in terms of `N', whereas a few with `n'. need to recheck again!

-All the simulation results of TFT need to compare with LSTM.

-What are the key factors influencing bike-sharing system activity in Hamburg, and how do these align with recent findings and trends globally?

-How does the performance of an interpretable TFT model compare with the existing state-of-the-art LSTM model in predicting bike-sharing system activity?

-Appendix A doen't connect well. need to put an effort to make this up to the mark.

-need to make conclusion concise.

-how does TFT model incremental compared to [63]?

-the expressed performance metrics are MAE, sMAPE, and RMSE, but unable to see any simulation results on them??? 

-need to provide clear statement on the contributions.

Author Response

We want to thank you for the valuable feedback and the time dedicated to reviewing our submitted paper.
Reviewer 6
ID Comment from the reviewer Reply from the authors
15 why does TFT model better than LSTM model? We have slightly expanded Sections 1 (Introduction) and 2 (Related Work) to incorporate our literature review, elucidating the novel practical benefits of the proposed model for operators, users, and policymakers when compared to the LSTM model. The addition of Table 7, accompanied by its preceding discussion, succinctly showcases the distinctive functionalities of the TFT model relative to the LSTM model that shows indications why it is performing better, while also acknowledging its drawback of requiring greater computational resources. Additionally, we have reorganized the conclusion section to commence with the model experiment to shift the immediate focus of readers to this topic.
16 many undefined parameters in each equation. A good hint. The undefined parameters have been clarified by an additional text.
17 provide a table for the notations. Done, this was also requested by Reviewer 1. We added this in the appendix A with a table.
18 Most of the equations have been expressed in terms of `N', whereas a few with `n'. need to recheck again! We have rechecked and adjusted them.
19 All the simulation results of TFT need to compare with LSTM. This is an interesting future field of work. In our study, however, we focus on the description of the TFT model predictions and the comparison with the expected patterns. Nevertheless, we have added further literature in the first section (Introduction) and the second section (Related Research), which deepen previous comparisons of different models (e.g., LSTM). In addition, we have added a comparative table (Table 7) in the Discussion, which also lists these models including our results.
22 Appendix A doen't connect well. need to put an effort to make this up to the mark. We are not exactly sure how the comment is meant. We have added an introductory sentence to clarify what can be seen in the appendix.
23 need to make conclusion concise. Done. The conclusion was quite extensive, as it also included the Outlook. We have split these two into two separate sections. Section 7 is now the Conclusion and therefore much more precise and concise, and Section 8 is the Outlook at the end of the article for the presentation of promising future research approaches.
24 how does TFT model incremental compared to [63]? We interpret the feedback from the review to suggest that we should address the extent of incremental improvements made to the Temporal Fusion Transformer (TFT) model. However, such improvements to the model's foundational architecture, as outlined in source [63], have not been undertaken by our team. Instead, our contribution lies in the application of the existing model to a novel context within the field of bike-sharing (exaptation). In reviewing our manuscript for areas requiring clarification, we observed that the paper consistently mentions and positions the models throughout its main body, explicitly detailing our approach and comparisons, for example:

- "As a result, our study introduces an interpretable attention-based Temporal Fusion Transformer (TFT) model, a novel approach in the realm of bike-sharing research. We compare its performance with the Long Short-Term Memory (LSTM) model, which is considered the state-of-the-art technology for predicting BSS activity." (1. Introduction)
- "Therefore, in our study, we employed LSTM as a baseline for comparison against our proposed model, an interpretable attention-based TFT." (3.7. Individual ML Models)
- "Table 6 presents the performance outcomes for both the TFT and LSTM models." (5.1. Performance Evaluation)
- "Furthermore, our study demonstrated that TFT, as a new proposed machine learning model in the realm of bike-sharing, significantly surpassed the performance of the existing state-of-the-art models while enabling full interpretability."  (6. Discussion)

There may be an expectation for our paper to detail incremental improvements to the TFT model. We made sure that there are no text parts that mention this and highlight its utilization in a novel application environment (BSS). Our research intentionally focuses on the original architecture as introduced by Lim et al. (https://doi.org/10.1016/j.ijforecast.2021.03.012). Our aim is to lay a groundwork and establish a foundational understanding, enabling future studies to build upon our findings and explore enhanced iterations of the TFT model's architecture.
25 the expressed performance metrics are MAE, sMAPE, and RMSE, but unable to see any simulation results on them???  The performance metrics were employed consistently during the development of the model and are detailed within the performance evaluation section, as illustrated in Table 6. Subsequently, we validated the plausibility of the model's predictions through the visualizations presented. However, it is important to note that these validations were not intended to serve as comprehensive simulations across all metrics. This decision was made in recognition that conducting such extensive simulations would surpass the scope of this paper, which represents the inaugural exploration of TFT models within the realm of bike-sharing systems. We eagerly anticipate future research that undertakes simulations utilizing TFT models across a broader spectrum of data metrics. 
26 need to provide clear statement on the contributions. Done. The most important contributions come right at the beginning and we hope that these are now clear and immediately apparent. To this end, the first sentences in Section 7 (Conclusion) directly state the two central contribution areas. We have also restructured the conclusion with regard to the concise presentation of the contributions of our article. 

Round 2

Reviewer 6 Report

Comments and Suggestions for Authors

Apart from my previous round review comments, authors have tried to address the comments. I am satisfied with authors response and revisions for few comments  but still I am not satisfied with many comments.  This revised version  is now much better,   but need to rework on the comments again. Ample scope is there to improve the response letter and revived version therein.

Comments on the Quality of English Language

Need to improve.